# Adeno-associated virus type 2 (AAV2) uncoating is a stepwise process and is linked to structural reorganization of the nucleolus

**Sereina O. Sutter**[1], **Anouk Lkharrazi**[1], **Elisabeth M. Schraner**[1], **Kevin Michaelsen**[1], **Anita Felicitas Meier**[1], **Jennifer Marx**[2], **Bernd Vogt**[1], **Hildegard Büning**[2], **Cornel Fraefel**[1] *

**1** Institute of Virology, University of Zurich, Zurich, Switzerland, **2** Institute of Experimental Hematology, Hannover Medical School, Hannover, Germany

☯ These authors contributed equally to this work.
\* cornel.fraefel@uzh.ch

**Data Availability Statement:** All relevant data are within the manuscript and its Supporting information files.

## Abstract

Nucleoli are membrane-less structures located within the nucleus and are known to be involved in many cellular functions, including stress response and cell cycle regulation. Besides, many viruses can employ the nucleolus or nucleolar proteins to promote different steps of their life cycle such as replication, transcription and assembly. While adeno-associated virus type 2 (AAV2) capsids have previously been reported to enter the host cell nucleus and accumulate in the nucleolus, both the role of the nucleolus in AAV2 infection, and the viral uncoating mechanism remain elusive. In all prior studies on AAV uncoating, viral capsids and viral genomes were not directly correlated on the single cell level, at least not in absence of a helper virus. To elucidate the properties of the nucleolus during AAV2 infection and to assess viral uncoating on a single cell level, we combined immunofluorescence analysis for detection of intact AAV2 capsids and capsid proteins with fluorescence *in situ* hybridization for detection of AAV2 genomes. The results of our experiments provide evidence that uncoating of AAV2 particles occurs in a stepwise process that is completed in the nucleolus and supported by alteration of the nucleolar structure.

## Author summary

Adeno-associated virus (AAV) capsids have been reported to enter the host cell nucleus and accumulate in the nucleolus. However, both the role of the nucleolus in AAV2 infection as well as the viral uncoating mechanism remain unknown. Here, we provide evidence that uncoating of the AAV2 particle is a stepwise process that is completed in the nucleolus and supported by alteration of the nucleolar morphology.

## Introduction

Adeno-associated virus type 2 (AAV2) is a small, non-pathogenic, helper virus-dependent parvovirus with a single-stranded (ss) DNA genome of approximately 4.7 kb. In absence of a

**Funding:** C.F. was supported by Swiss National Science Foundation No. 310030_184766 (https://www.snf.ch/en). The funders had no role in study design, data collection and analysis, decision to publish, or preparation of the manuscript.

**Competing interests:** The authors have declared that no competing interests exist.

helper virus, AAV2 can integrate its genome site-preferentially into the adeno-associated virus integration site 1 (AAVS1) on human chromosome 19 or persist in an episomal form in the nucleus [1,2]. Co-infection with a helper virus, such as herpes simplex virus type 1 (HSV-1), leads to a lytic replication cycle including the production of progeny virus particles [3]. The AAV2 genome consists of two large open reading frames (ORFs) flanked by 145 nt long inverted terminal repeats (ITRs) located on either side. The *rep* gene encodes the four non-structural Rep proteins, two of which are transcribed from the p5 and the p19 promoter, respectively. An alternative splice site regulates expression of the alternative transcripts, whereby the unspliced RNAs encode Rep78 and Rep52, whereas Rep68 and Rep40 are encoded by their corresponding spliced variant [4,5]. The promoter activity is regulated by the Rep binding site (RBS), therefore allowing the Rep protein to act either as a trans-activator or repressor [6]. In the absence of a helper virus, only little expression of Rep takes place, which nonetheless is sufficient to repress any further transcription.

The three structural proteins VP1, VP2 and VP3, constituting the icosahedral capsid, are encoded by the *cap* gene. Furthermore, the *cap* gene encodes the assembly-activating protein (AAP) and the membrane-associated accessory protein (MAAP) by means of nested alternative ORFs [7,8].

Adeno-associated viruses exhibit a broad cellular tropism [9]. Referring to AAV2, the cellular receptors facilitating cell attachment and entry, include heparan sulfate proteoglycan, human fibroblast growth factor receptor 1, $\alpha_V\beta_5$ integrin, $\alpha_5\beta_1$ integrin (reviewed in [10]) and the host factor KIAA0319L (synonymous AAVR) [11]. Different entry pathways were proposed for AAV2, including clathrin- and dynamin-dependent endocytosis or internalization supported by the Ras-related C3 botulinum toxin substrate 1 (Rac1), a small GTPase and a major effector of macropinocytosis (reviewed in [10]). However, internalization through clathrin-independent carriers (CLICs) and GPI-enriched endocytic compartments (GEECs) was also reported as major endocytic infection route [12]. It was shown that acidification in endocytic compartments and the activity of proteases trigger conformational changes of the AAV2 capsid, leading to the exposure of the N-terminal domain of the VP1 protein, known as VP1 unique region ($VP1_u$). $VP1_u$ containing a phospholipase A2 domain ($PLA_2$) as well as a nuclear localization signal enables the endosomal escape of AAV2 and nuclear entry, respectively [13]. After nuclear entry, AAV2 capsids were shown to accumulate in nucleoli [14], possibly mediated by nucleoli-associated proteins [15,16]. Nucleoli are membrane-less structures located within the nucleus and are organized in three distinct compartments. The fibrillary center is surrounded by the dense fibrillary compartment and further embedded in the granular compartment. The structural (re-)organization of the nucleolus is strongly linked to its function in transcription and pre-rRNA processing. Besides, the nucleolus is known to be involved in many other cellular functions, including stress response, cell cycle regulation and apoptosis (reviewed in [17–20]). Additionally, many different viruses such as HSV-1, human immunodeficiency virus type 1 (HIV-1) or adenovirus (AdV) can harness the nucleolus or specific nucleolar proteins in order to promote different steps of their life cycle including replication, transcription and virus assembly [21,22]. While AAV2 capsids have previously been reported to enter the host cell nucleus and accumulate in the nucleolus, probably in a nucleolin- and nucleophosmin-dependent manner [15,16], both the role of the nucleolus in AAV2 infection, and the viral uncoating mechanism remain elusive. The current paradigm proposes a model where AAV2 capsids accumulate in the nucleolus upon nuclear entry and then translocate to the nucleoplasm for uncoating [14]. This model is based on the observation that treating cells with either proteasome inhibitors or hydroxyurea, both known to enhance AAV2 transduction, improve nucleolar accumulation and mobilization of virions into the nucleoplasm, respectively. Besides, the post-transcriptional silencing of nucleophosmin, a highly

expressed multifunctional nucleolar phosphoprotein, enhanced nucleolar accumulation and increased transduction similar to the treatment with proteasome inhibitors, while silencing of nucleolin, an abundant non-ribosomal protein of the nucleolus [23], mobilized capsids to the nucleoplasm and enhanced transduction similar to the treatment with hydroxyurea. Other studies concluded that uncoating occurs before or during nuclear entry [24–26]. These conclusions were mainly based on the fact that only AAV2 genomes were detected in the nucleus and that perinuclear genomes did not always co-localize with AAV2 capsids. In all these prior studies on AAV uncoating, however, viral capsids and viral genomes were not directly correlated on the single cell level, at least not in absence of a helper virus [27], but rather quantified by quantitative (q)PCR, Western or slot blot analysis, single AAV capsid specific immunostainings or fluorescently labelled AAV virions [14,24–30].

To elucidate the sink-like properties of the nucleolus during AAV2 infection and to assess viral uncoating on a single cell level, we combined immunofluorescence (IF) analysis to detect intact AAV2 capsids as well as capsid proteins with fluorescence *in situ* hybridization (FISH) to visualize AAV2 genomes. The results of our experiments support the hypothesis that AAV2 uncoating takes place in the nucleolus in a cell cycle-dependent manner.

## Results

### AAV2 capsids and AAV2 genomes accumulate in the nucleoli of infected cells

Previous studies have shown that nucleoli act as a sink for incoming AAV2 capsids. However, it is unknown whether the nucleolar localization is merely a result of the interaction of the AAV2 capsids with specific nucleolar proteins or a pre-requisite for an early step of the viral replication cycle such as uncoating or second strand-synthesis. While prior reports suggested that AAV2 capsids translocate from the nucleoli to the nucleoplasm for uncoating, these studies have not simultaneously tracked capsids and genomes on the single cell level [14,31].

Here, we investigated the spatial and temporal distribution of both AAV2 capsids and AAV2 genomes in single cells. For this, normal human fibroblast (NHF) cells were either mock-infected or infected with wild-type (wt) AAV2 at a multiplicity of infection (MOI) of 20'000 genome containing particles (gcp) per cell (herein referred to as MOI). The cells were fixed at different time points post infection and processed for combined multicolor immunofluorescence (IF) analysis to detect AAV2 particles using an antibody that detects a conformational capsid epitope and fluorescence *in situ* hybridization (FISH) to detect AAV2 genomes (hereinafter referred to as IF-FISH). The results showed both AAV2 capsids and AAV2 DNA accumulated in the nucleoli of wtAAV2 infected cells over time (Fig 1A and 1B). The nucleolar accumulation of AAV2 capsids and AAV2 DNA was observed also at a ten-fold lower MOI both in NHF cells and lung epithelial (A549) cells (S1 Fig) and with recombinant AAV2 vectors (S2 Fig). Of note, transduction efficiency (% GFP positive cells) correlated with the uncoating efficiency (% rAAV DNA-positive and rAAV capsid-negative cells).

Neither AAV2 capsids nor genomes were detected in the nucleoli upon infection of cells with mutant AAV2 [76]HD/AN (S3 Fig) which contains two mutated residues in the catalytic center of the phospholipase A2 (PLA$_2$) domain and is therefore deficient for endosomal escape [32].

Interestingly, we did not only observe capsid-positive, genome-negative (AAV2 capsid +DNA-) signals in the cytoplasm (cytosol and the endocytic organelles contained therein), as it would be expected when capsids are intact and therefore do not allow binding of the FISH probe to the virus genome, but frequently also capsid-positive, genome-positive signals (AAV2 capsid+DNA+; see insets in Fig 1A, 0 h, 3 h, 10 h and Fig 1C and 1D), which largely vanished

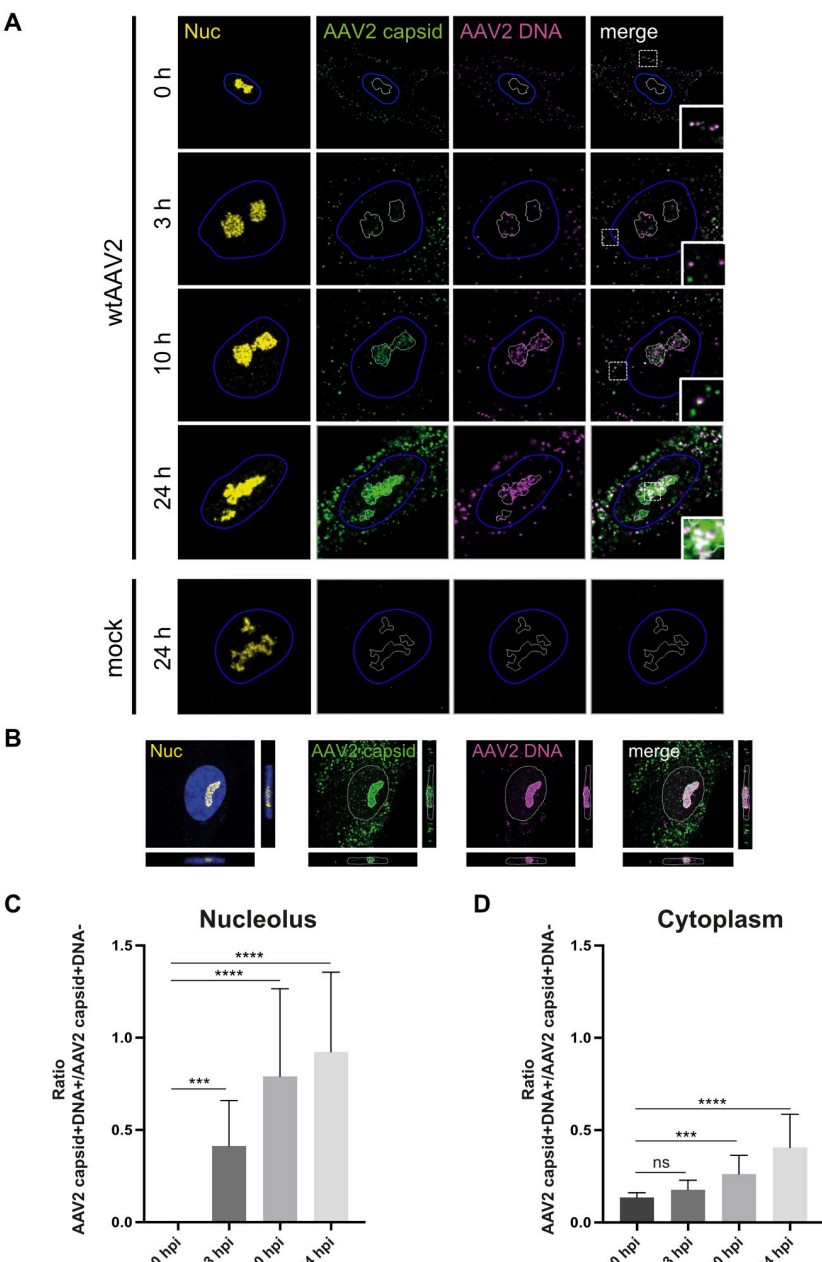

**Fig 1. Spatial distribution of AAV2 capsids and DNA over time.** NHF cells were infected with wtAAV2 (MOI 20'000). At various time points post infection, the cells were fixed and processed for multicolor IF analysis combined with FISH and CLSM. Nucleoli (Nuc) were visualized using an antibody against fibrillarin (yellow). Intact capsids were stained using an antibody that detects a conformational capsid epitope (green). AAV2 DNA (magenta) was detected with an Alexa Fluor (AF) 647 labeled, amine-modified DNA probe that binds to the AAV2 genome. (A) Spatial and temporal distribution of AAV2 capsids and DNA. The white line represents the edge of the nucleoli (fibrillarin staining), the blue line represents the edge of the nucleus (4',6-diamidino-2-phenylindole (DAPI) staining). (B) Orthogonal projections of a z-stack at 24 hpi. The white line represents the edge of the nucleus (DAPI staining). (C) Image-based quantification of the uncoating rate of 50 individual cells per time point in the nucleolus and (D) in the cytoplasm. p-values were calculated using an unpaired Student's t-test (*—p ≤ 0.05, **—p ≤ 0.01, ***—p ≤ 0.001, ****—p ≤ 0.0001).

after DNase I treatment (S4 Fig). This indicates that either the virus stocks contained improperly encapsidated AAV2 DNA or that the AAV2 genome is accessible to the FISH probe within the cytoplasm (herein referred to as genome accessibility). While the main focus of this study was on simultaneously tracking AAV2 capsids and AAV2 genomes in the nucleus, it was important to first investigate the origin of the AAV2 capsid-positive and AAV2 genome-positive signals in the cytoplasm (Fig 1A insets and Fig 1D).

## Co-detection of AAV2 capsids and AAV2 genomes in the cytoplasm is supported by AAV2 genome accessibility and requires acidification

To address the question whether wtAAV2 stocks contained improperly encapsidated AAV2 genomes, wtAAV2 particles were directly applied to fibronectin coated coverslips and processed for IF-FISH (Fig 2A). While in the untreated wtAAV2 samples all capsid-positive signals (green) were genome-negative, only genome-positive signals (red) but no capsids were observed upon incubation for 5 min at 75°C, which is known to destabilize AAV2 capsids [33]. In the heat-treated samples, the AAV2 capsids were indeed disintegrated as confirmed by electron microscopy (Fig 2B), and the AAV2 genome signals disappeared upon DNase I

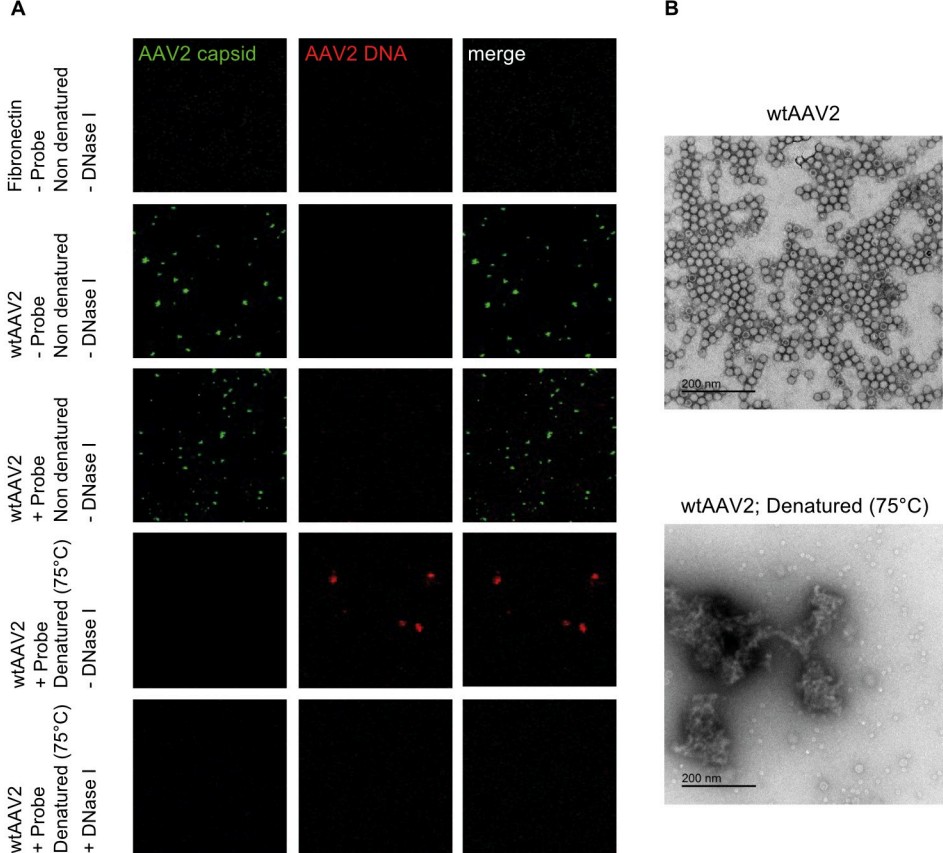

**Fig 2. AAV2 particles on fibronectin coated coverslips.** (A) wtAAV2 particles were directly applied to fibronectin coated coverslips and processed for IF analysis combined with FISH and CLSM, exactly as described for Fig 1. Intact capsids were stained using an antibody that detects a conformational capsid epitope (green). AAV2 DNA (red) was detected with an Alexa Fluor (AF) 647 labeled, amine-modified DNA probe that binds to the AAV2 genome. (B) Electron photomicrographs show complete disintegration of the AAV2 capsids at 75°C.

treatment (Fig 2A). These experiments demonstrate that the virus stocks were not contaminated with improperly encapsidated AAV2 DNA. To address the hypothesis that co-detection of AAV2 capsids and AAV2 genomes in the cytoplasm is enabled by genome accessibility, we determined the ratios of AAV2 capsid+DNA+/AAV2 capsid+DNA- signals at different time-points after infection using CellProfiler. As shown in Fig 1C and 1D, these ratios significantly increased with time of infection, both in the cytoplasm and the nucleoli, indicating that AAV2 capsid+DNA- signals decrease over time.

As acidification has been shown to lead to conformational changes in the AAV2 capsid and to be important for AAV2 infection, endosomal escape, and nuclear entry in particular [25], we examined whether acidification leads to co-detection of AAV2 capsid- and AAV2 genome signals in the cytoplasm, likely within endocytic vesicles of this compartment [34]. To this end, NHF cells were treated with bafilomycine A1 (50 or 200 nM) 1 h prior to infection with wtAAV2 (MOI 20'000). At 3 hours post infection (hpi), the samples were processed for IF-FISH and CLSM. Treating cells with bafilomycine A1, a vacuolar H+-ATPase inhibitor which blocks endosomal acidification, significantly reduced the AAV2 capsid+DNA+/AAV2 capsid+DNA- signal ratios in the cytoplasm (Fig 3A and 3B; see also insets in Fig 3A) and, as demonstrated previously [25], also the import of AAV2 capsids into the nucleus (Fig 3C). Collectively, these experiments confirm the specificity of the IF-FISH assay and support the hypothesis that the co-localization of AAV2 capsids and AAV2 DNA in the cytoplasm is due to genome accessibility and is enhanced by acidification.

## Complete AAV2 uncoating takes place in the nucleoli

After establishing that the co-detection of AAV2 capsids and AAV2 genomes by combined IF-FISH and CLSM is part of the infection biology and not caused by improperly encapsidated virus particles, we continued to analyze the distribution of AAV2 capsids and genomes in the nuclei of individual cells. For this, NHF cells were mock-infected or infected with wtAAV2 (MOI 20'000) and 24 h later processed for combined IF-FISH and CLSM to detect AAV2 capsids and genomes. Interestingly, we observed three distinct patterns of nucleolar AAV2 genome and AAV2 capsid staining: (I) nucleoli with robust AAV2 genome and AAV2 capsid signal, (II) nucleoli with robust AAV2 DNA signal but weak AAV2 capsid signal, and (III) nucleoli in which only the viral DNA was detected in absence of capsids (Fig 4; see also S1 Movie). The pattern, in particular the observation of AAV2 DNA in the nucleoli in absence of AAV2 capsids, led to the hypothesis that complete AAV2 uncoating takes place in the nucleolus.

In order to assess the functionality of the nucleolar AAV2 DNA, NHF cells were either infected with wtAAV2 or rAAVCFPRep (a recombinant AAV2 vector encoding the cyan fluorescent reporter protein fused in frame with AAV2 Rep under the control of the wt p5 promoter; both at a MOI 20'000). After 24 h, the cells were fractionated (S5 Fig), and purity of the nuclear, nucleolar, and nucleoplasmic fractions were confirmed by IF-FISH analysis and Western blot. Transfection of bead-purified (uncoated) nucleolar rAAVCFPRep DNA resulted in CFP-signal in the transfected cells (S6 Fig), indicating that the rAAV DNA released into the nucleoli is transcriptionally active. Southern analysis of the nucleolar fraction prepared at 24 and 48 hpi revealed bands representing the single-stranded (ss)AAV2 DNA and, most interestingly, the replication form monomer (rfm) which is the product of AAV2 second-strand synthesis (Fig 5). In the nucleoplasmic fraction neither ssAAV2 DNA nor rfm was detected at 24 hpi. At 48 hpi, a band representing the ssAAV2 DNA became visible also in the nucleoplasmic fraction, however, although the signal intensity of that band was comparable to the respective

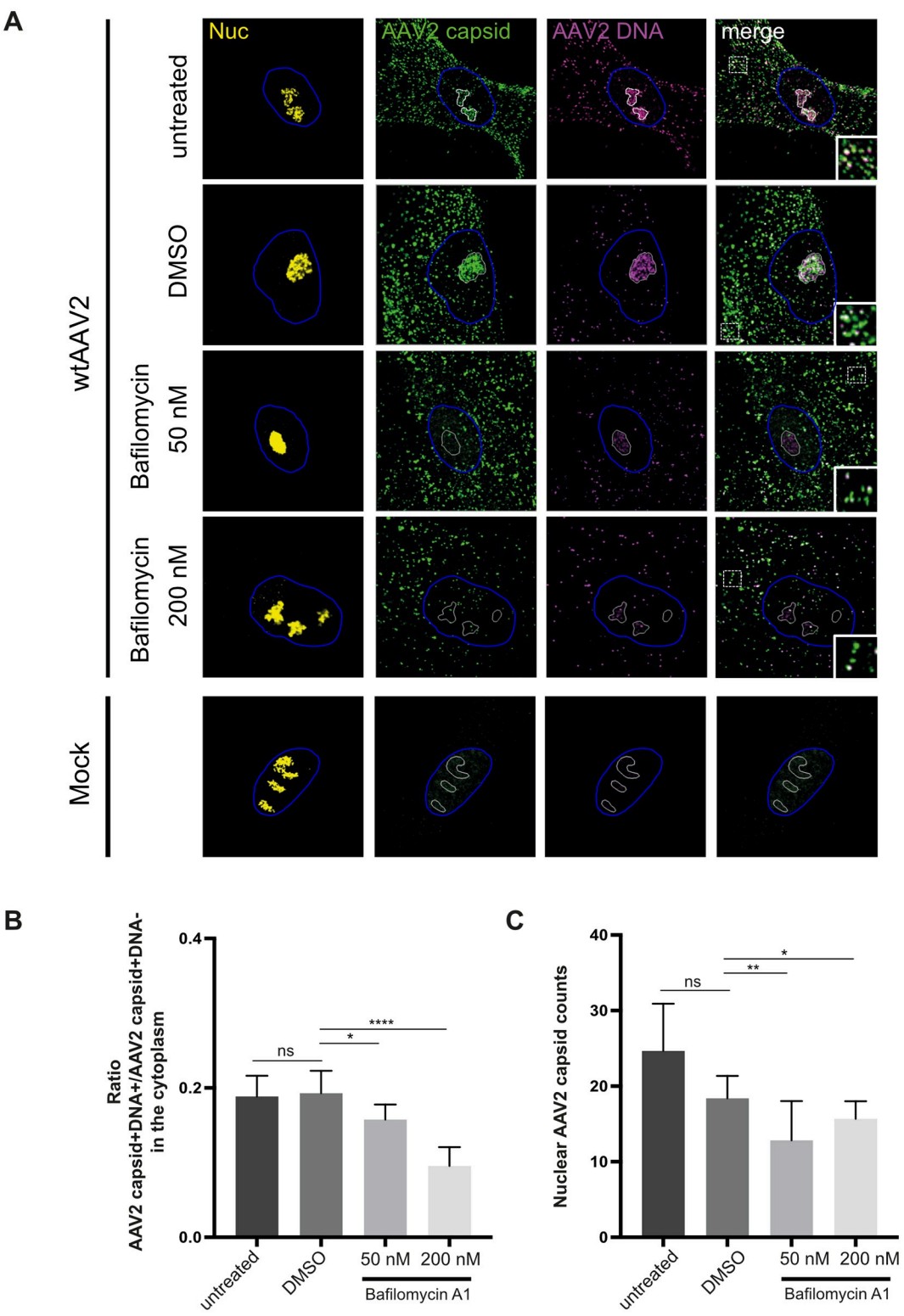

**Fig 3. Acidification enhances genome accessibility of AAV2.** NHF cells were treated with bafilomycine A1 (50 or 200 nM) or DMSO 1 h prior to infection with wtAAV2 (MOI 20'000). At 3 hpi, the cells were fixed and processed for multicolor IF analysis combined with FISH and CLSM. Nucleoli (Nuc) were visualized using an antibody against fibrillarin (yellow). Intact capsids were stained using an antibody that detects a conformational capsid epitope (green). AAV2 DNA (magenta) was detected with an Alexa Fluor (AF) 647 labeled, amine-modified DNA probe that binds to the AAV2 genome. (A) Genome

accessibility of AAV2 capsids after inhibition of the acidification of the endosome-lysosome system. The white line represents the edge of the nucleoli (fibrillarin staining), the blue line represents the edge of the nucleus (DAPI staining). (B) Image-based quantification of the genome accessibility (ratio of AAV2 capsid+DNA+/AAV2 capsid+DNA- signal) of 50 individual cells per sample in the cytoplasm and (C) nuclear AAV2 capsid counts. p-values were calculated using an unpaired Student's t-test (*—p ≤ 0.05, **—p ≤ 0.01, ***—p ≤ 0.001, ****—p ≤ 0.0001).

band observed in the nucleolar fraction, no rfm was detected in the nucleoplasmic fraction by 48 hpi.

## Detection of AAV2 capsid proteins in absence of AAV2 capsids

If the absence of AAV2 capsid staining in the nucleoli with positive AAV2 genome signal was indeed due to complete viral uncoating, we would expect the presence of disassembled AAV2 capsid proteins in those nucleoli. To assess this hypothesis, NHF cells were mock-infected or infected with wtAAV2 (MOI 20'000) and 24 h later processed for combined IF-FISH and CLSM to detect AAV2 capsids (conformational epitope), AAV2 capsid proteins (linear epitope) and AAV2 DNA (Figs 6 and S7). The results show the absence of AAV2 capsids in presence of AAV2 capsid proteins in the nucleolus, supporting the hypothesis that AAV2 uncoating indeed takes place in the nucleoli. For technical reasons, co-staining of AAV2 capsids, AAV2 capsid proteins VP1/2/3 and AAV2 DNA did not allow to directly visualize nucleoli. However, the DAPI staining in Fig 6 indirectly reveals the position of the nucleoli, since the nucleoli have a lower DNA density than the surrounding structures and therefore appear as dark regions [35]. Moreover, individual staining of either AAV2 capsids or AAV2 capsid proteins together with AAV2 DNA and a nucleolar marker demonstrated that both intact AAV2 capsids (e.g., Fig 1) and AAV2 capsid proteins VP1/2 (Fig 6B) accumulated with AAV2 DNA in nucleoli.

Intriguingly, we noticed a distinct difference in the nucleolar structure when comparing AAV2 DNA-positive nucleoli that were positive also for intact AAV2 capsids with AAV2 DNA-positive nucleoli that were negative for intact AAV2 capsids or positive for AAV2 capsid proteins. Specifically, the nucleoli appeared dense when positive for both AAV2 DNA and AAV2 capsids and dispersed when positive for AAV2 DNA and negative for AAV2 capsids or positive for AAV2 DNA and AAV2 capsid proteins (Figs 4A and 6B). The image-based quantification of the mean integrated intensity of AAV2 capsid signals relative to the nucleolar structure revealed a higher capsid signal intensity in dense nucleoli than in dispersed nucleoli (Fig 4C). Overall, these experiments imply that complete AAV2 uncoating takes place in the nucleoli and coincides with changes in the nucleolar structure.

As the ratios of dense to dispersed nucleoli was comparable in mock-infected and wtAAV2 infected cells (S8 Fig), we hypothesized that not virus infection *per se* but cellular processes such as apoptosis, stress response, or cell cycle regulation are responsible for the different structures of AAV2 capsid-positive and AAV2 DNA-positive versus AAV2 capsid-negative and AAV2 DNA-positive nucleoli and may thereby control AAV2 uncoating.

## Changes in nucleolar morphology correlate with cell cycle progression

To address the question whether the changes in nucleolar structure are linked to cell cycle progression, we performed image-based cell cycle analysis using DAPI and fibrillarin staining. To this end, a DAPI integrated intensity protocol was adapted from Ruokos et al. [36] and validated in NHF cells by correlating cyclin A, which is only expressed in late S and G2 cell cycle

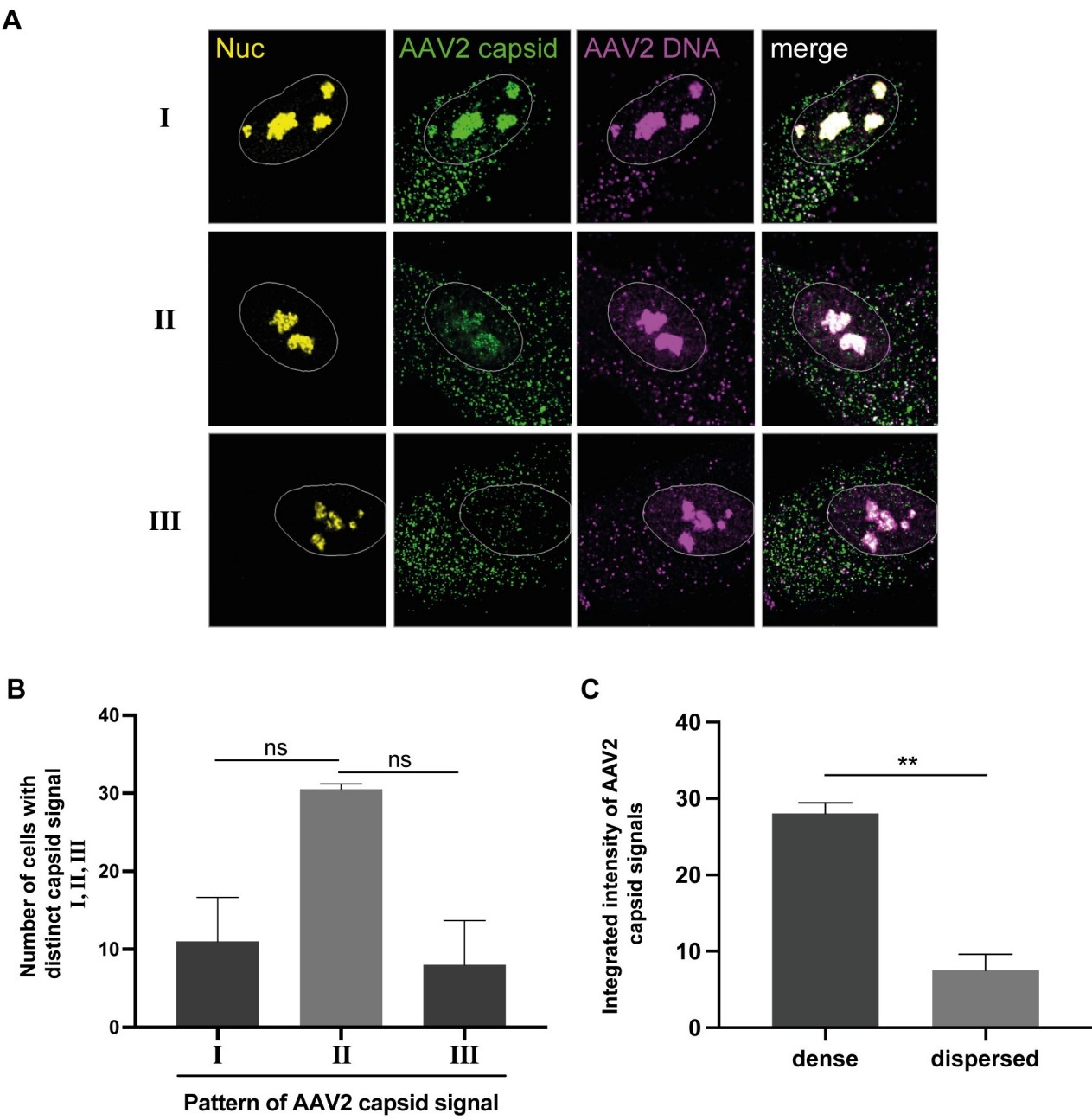

**Fig 4. Absence of intact AAV2 capsids in AAV2 genome positive nucleoli points towards complete viral uncoating.** NHF cells were infected with wtAAV2 (MOI 20'000). At 24 hpi, the cells were fixed and processed for multicolor IF analysis combined with FISH and CLSM. Nucleoli (Nuc) were visualized using an antibody against fibrillarin (yellow). Intact capsids were stained using an antibody that detects a conformational capsid epitope (green). AAV2 DNA (magenta) was detected with an Alexa Fluor (AF) 647 labeled, amine-modified DNA probe that binds to the AAV2 genome. Nuclei were counterstained with DAPI and illustrated as white lines. (A) Distinct pattern (I—III) of AAV2 capsid signal in cells with AAV2 genome positive nucleoli. (B) Quantification of 50 individual cells with distinct AAV2 capsid signal. (C) Image-based quantification of the integrated intensity of AAV2 capsid signals in dense or dispersed nucleoli of 70 individual cells. p-values were calculated using an unpaired Student's t-test ($*$—$p \leq 0.05$, $**$—$p \leq 0.01$, $***$—$p \leq 0.001$, $****$—$p \leq 0.0001$).

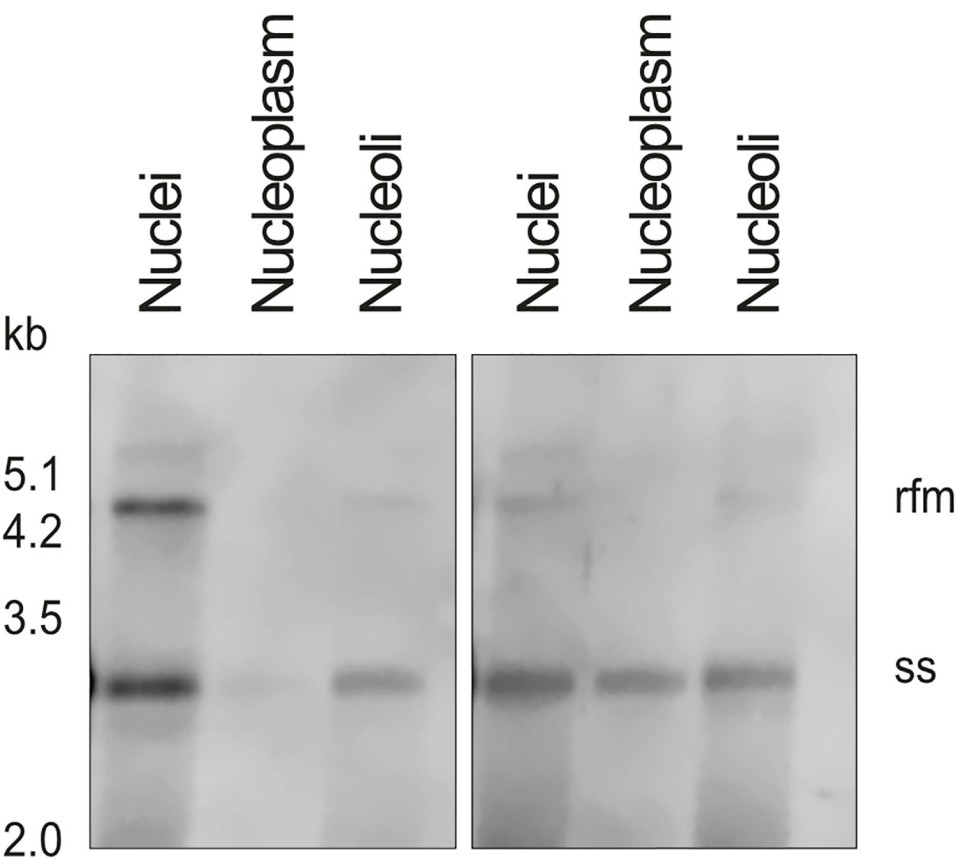

**Fig 5. Southern analysis of AAV2 DNA in cell fractions.** Southern Blot of denatured Hirt DNA extracted at 24 and 48 hpi from isolated nuclei, nucleoplasm, or nucleoli of NHF cells infected with wtAAV2 (MOI 20'000; 1.5x10^6 cell equivalents per lane). AAV2 DNA was visualized with an AAV2 *rep*-specific probe disclosing bands representing the single-stranded (ss)AAV2 DNA and the replication form monomer (rfm; product of AAV2 second-strand synthesis). Representative images from two independent experiments are shown.

phases, with the integrated intensity of DAPI (S9 Fig; see also Materials and methods). As a first step, the background of each image was subtracted (step 1). Next, nuclei as well as the cyclin A staining were identified as primary objects using CellProfiler (step 2). In step 3 and 4, nuclei and cyclin A were related to each other and the DAPI integrated intensity of each cell was measured. The measured properties of each individual cell were subsequently exported and read into Matlab, where histograms were plotted. Visual thresholds were set (red dotted lines) to distinguish the distribution of the histogram into G1, S and G2 (step 5). The images were then analyzed with a second CellProfiler pipeline using the visual thresholds of the integrated intensity of DAPI to classify cells into G1: 54.85%, S: 9.67% and G2: 35.48% (step 6). Lastly, overlay images, showing the cell cycle stage of each cell, were saved to allow the tracking of individual cells for further analysis (step 7).

To further validate the adapted protocol, NHF cells were synchronized using a double thymidine block. After the release, the cells were either mock-treated or treated with nocodazole (200 nM) for 24 hours to induce a G2 cell cycle arrest. For flow cytometry, the cells were harvested, fixed and stained with DAPI. For CLSM, the coverslips were embedded in ProLong Anti-Fade mountant containing DAPI. Images were analyzed as described in the section cell cycle analysis based on 4',6-diamidino-2-phenylindole (DAPI) staining. Both methods, flow

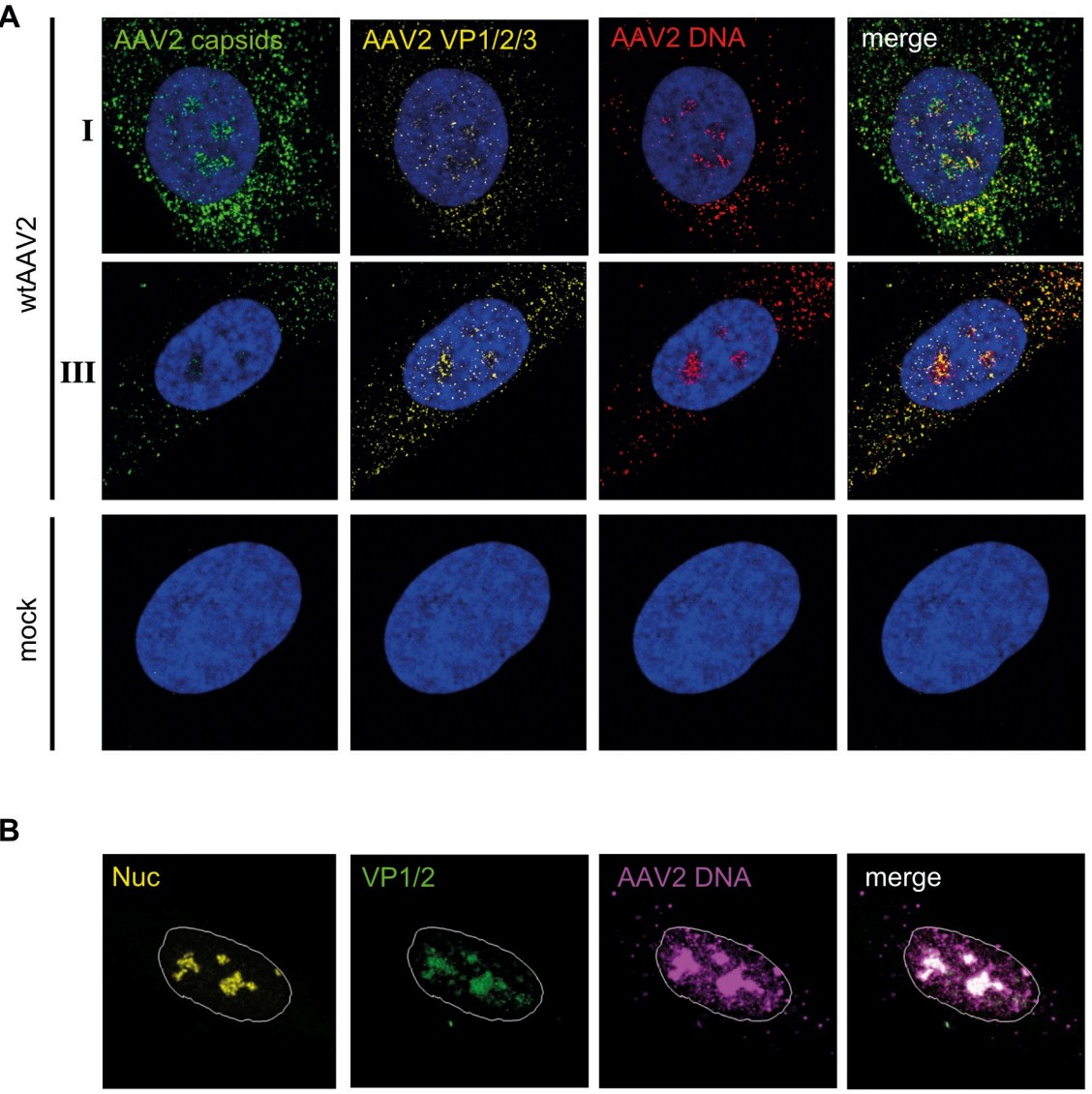

**Fig 6. Co-detection of AAV2 DNA with AAV2 capsids and AAV2 capsid proteins.** NHF cells were infected with wtAAV2 (MOI 20'000). At 24 hpi, the cells were fixed and processed for multicolor IF analysis combined with FISH and CLSM. (A) Intact capsids (green) or capsid proteins (yellow) were detected using either an antibody against intact AAV2 capsids (conformational capsid epitope) or an antibody (linear epitope) against VP1, VP2 and VP3. AAV2 DNA (red) was detected with an Alexa Fluor (AF) 647 labeled, amine-modified DNA probe that binds to the AAV2 genome. Nuclei were counterstained with DAPI (blue). The wtAAV2 infected cells shown were assigned as pattern I and III according to the intensity of the capsid staining and following a similar classification as defined in Fig 4. (B) AAV2 capsid proteins (green) were detected using an antibody (linear epitope) against VP1 and VP2. AAV2 DNA (magenta) was detected by linking the amine-modified DNA to AF647. Nucleoli (Nuc) were visualized using an antibody against fibrillarin (yellow). Nuclei were counterstained with DAPI and illustrated as white lines.

cytometry and CLSM, showed a significant decrease of the number of cells within G1 cell cycle phase and a significant increase of cells in G2 cell cycle phase upon nocodazole treatment (S10 Fig), indicating that the adapted protocol is suitable for image-based cell cycle staging. Next, the fibrillarin staining was correlated to the cell cycle profile of the mock-infected and wtAAV2 infected NHF cells. Fig 7 shows that the ratios of dense to dispersed nucleoli decrease

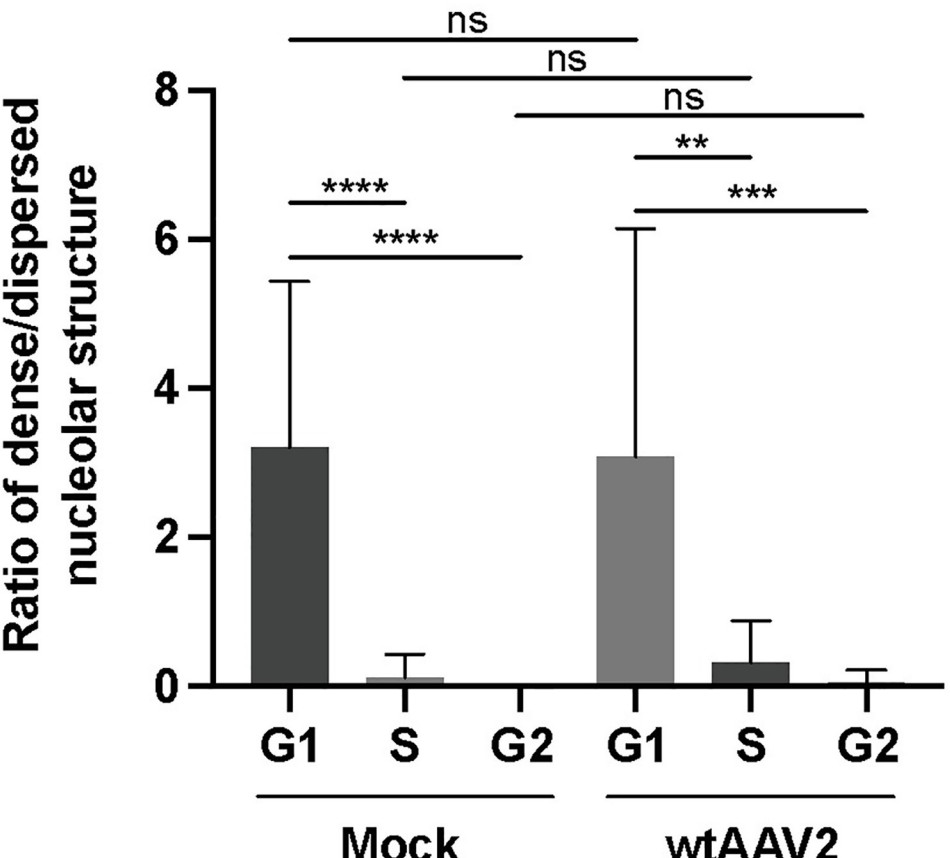

**Fig 7. Nucleolar reorganization during cell cycle progression.** Image-based analysis of the ratios of dense to dispersed nucleoli of 100 individual mock- or AAV2 infected cells in different cell cycle phases (G1, S and G2). p-values were calculated using an unpaired Student's t-test (*—p ≤ 0.05, **—p ≤ 0.01, ***—p ≤ 0.001, ****—p ≤ 0.0001).

during cell cycle progression in both mock-infected and AAV2 infected cells. Overall, the data imply that the observed morphological changes of the nucleoli indeed coincide with cell cycle progression and were not due to AAV2-induced stress response (see also S8 Fig).

## G1 cell cycle phase obstructs complete AAV2 uncoating in nucleoli

Since we observed a decrease in the ratio of dense versus dispersed nucleolar structures (Figs 7 and S11) during cell cycle progression and a stronger intensity of AAV2 capsid signals in dense nucleoli than in dispersed nucleoli (Fig 4C), we next addressed the question whether cell cycle progression is important for complete AAV2 uncoating. For this, NHF cells were arrested in the G1 phase by a double thymidine treatment before and during infection with wtAAV2 (MOI 20'000). As control, the cells were released 8 h prior to infection by washing out the thymidine. At 24 h after infection, there was a robust difference in the cell cycle profile between G1-arrested cells (approx. 83% in G1) and released cells (approx. 47% in G1), confirming the efficient G1 arrest (Fig 8A). Image-based cell cycle analysis showed that the rate of complete uncoating (AAV2 capsid-DNA+/AAV2 capsid+DNA+) in nucleoli was

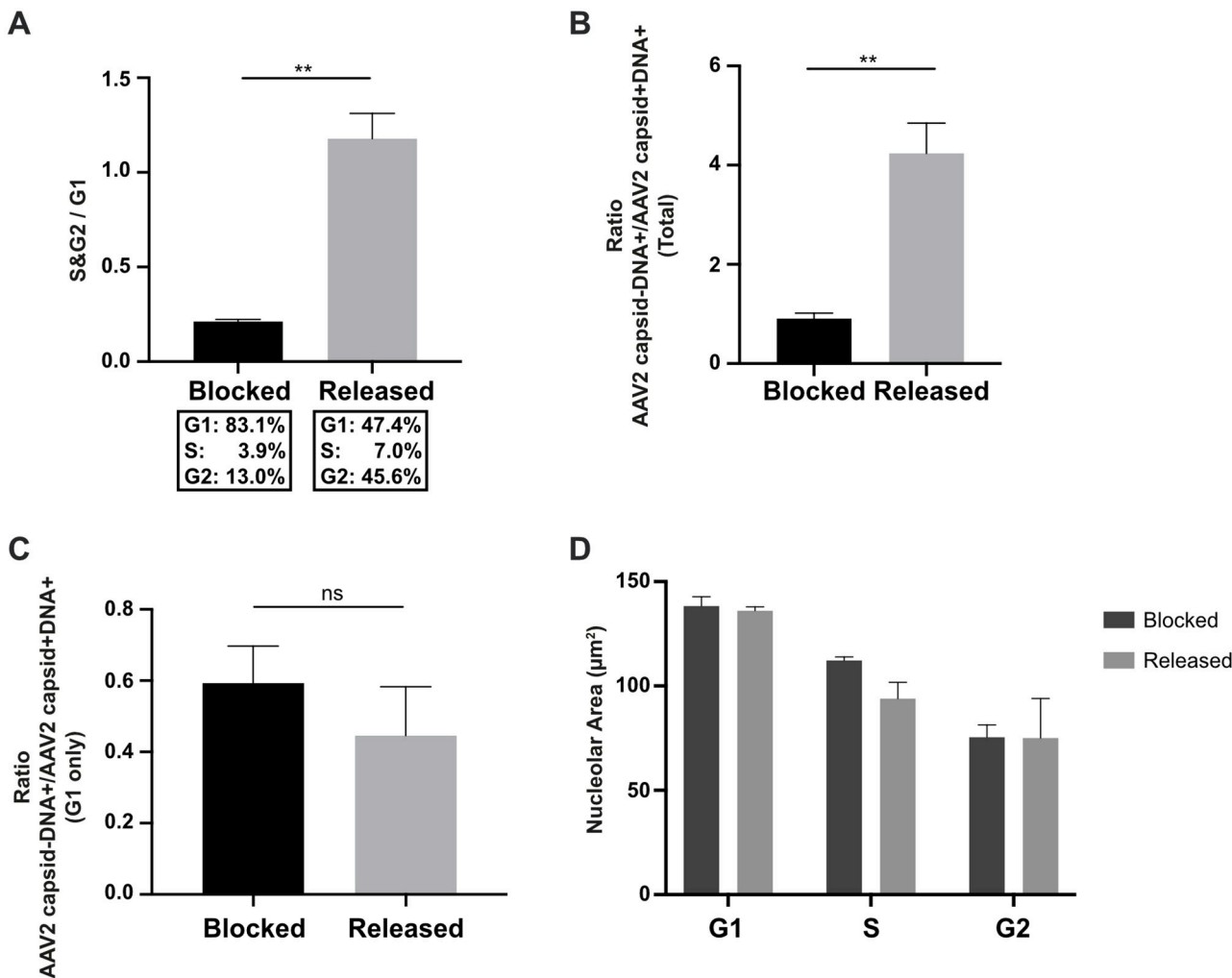

**Fig 8. G1 cell cycle arrest obstructs complete uncoating in nucleoli.** NHF cells were arrested in G1 cell cycle phase by a double thymidine block before and during infection with wtAAV2 (MOI 20'000). As control, the cells were released 8 h prior to infection by washing out the thymidine. At 24 hpi, the cells were fixed and processed for multicolor IF analysis combined with FISH, CLSM and image-based analysis of (A) the cell cycle profile after continuous thymidine block or release, respectively. (B) Quantification of the total uncoating rate in nucleoli. (C) Image-based quantification of the uncoating rate in nucleoli in G1 cell cycle phase. (D) Image-based quantification of the nucleolar area after continuous thymidine block or release, respectively. p-values were calculated using an unpaired Student's t-test (*—p ≤ 0.05, **—p ≤ 0.01, ***—p ≤ 0.001, ****—p ≤ 0.0001).

approximately 4-fold lower in the G1-arrested cells compared to the released cells (Fig 8B). The double thymidine block did not *per se* influence the rate of complete uncoating in nucleoli, as the ratios of AAV2 capsid-DNA+/AAV2 capsid+DNA+ signals in G1 cells were comparable in presence or absence of thymidine (Fig 8C). Moreover, neither the blocking with nor the release from thymidine influenced the total area of the nucleoli during cell cycle progression (Fig 8D).

Similar to the effect of the thymidine treatment, contact inhibited NHF cells showed a defect in the rate of complete uncoating (AAV2 capsid-DNA+/AAV2 capsid+DNA+) in nucleoli compared to cycling cells (S12A Fig). Of note, in the cycling cells the numbers of cells with nucleolar AAV2 capsid-positive and DNA-positive (AAV2 capsid+DNA+) signal increased from 0–10 hpi (likely due to accumulation of incoming capsids) and then decreased

from 10–48 hpi (S12B Fig), while the number of cells in which the nucleolar AAV2 DNA accumulated together AAV2 capsid proteins VP1, VP2 and VP3 (AAV2 VP$_{1/2/3}$+DNA+) increased from 0–48 hpi (S12C Fig).

## Induction of nucleolar disruption overcomes thymidine-mediated obstruction of AAV2 uncoating

NHF cells were arrested in the G1 phase of the cell cycle by a double thymidine block before and during infection with wtAAV2 (MOI 20'000). At 24 hpi, the cells were treated with actinomycin D (50 nM) for 1 h in order to induce nucleolar disruption and nucleolar cap formation (reviewed in [37]), fixed and processed for multicolor IF-FISH and CLSM (Fig 9A). Analysis of the cell cycle profile confirmed the cell cycle arrest upon thymidine and actinomycin D treatment (60% of cells in G1). The actinomycin D mediated nucleolar disruption in thymidine-treated cells led to a considerable decrease of capsid signals in the nucleoli and nucleoplasm and an increase of AAV2 genome signals in the nucleoplasm (Fig 9A and 9B). This shows that complete uncoating in nucleoli can be induced by changes in the nucleolar structure (disruption) even when cells are in G1 phase where normally no efficient complete uncoating is observed (Fig 8).

## Capsid disassembly coincides with cell cycle progression

To further assess whether capsid disassembly overlaps with cell cycle progression, NHF cells were either mock-infected or infected with wtAAV2 (MOI 20'000). 24 h later, the cells were fixed and processed for IF-FISH, CLSM and image-based cell cycle analysis and quantification. Specifically, we used the DAPI integrated intensity protocol to determine the cell cycle phase and the IF-FISH protocol to detect intact AAV2 capsids, the disassembled AAV2 VP1/2/3 capsid proteins, and the AAV2 DNA (Fig 10). In 55% of the cells in G1 cell cycle phase but only in 29% of the cells in S/G2 we observed the accumulation of AAV2 DNA together with AAV2 capsids (genome accessibility, pattern I). In contrast, the number of cells in which the AAV2 DNA did not accumulate together with AAV2 capsids (pattern III) but rather with AAV2 capsid proteins VP1, VP2 and VP3 (complete uncoating) increased from 50% in G1 to 80% in S/G2. The same observation held true for neonatal human dermal fibroblasts (HDFn) cells infected with wtAAV2 (S13 Fig). Overall, our data strongly indicate that capsid disassembly coincides with cell cycle progression and nucleolar alterations.

## Discussion

Nucleoli are membrane-less and dynamic subnuclear structures, which were mainly known for their role in ribosome biosynthesis. However, nucleoli have a function in numerous other cellular processes as well, such as cell cycle regulation, stress response and apoptosis (reviewed in [17–20]). Proteomic approaches led to the identification of roughly 4'500 nucleolar associated proteins of which only a third is linked to ribosome biogenesis [38,39].

Many different viruses can exploit the nucleolus or nucleolar proteins to drive different steps of their life cycle including replication, transcription, and assembly (reviewed in [21,22,40,41]). For example, HSV-1 induces the redistribution of nucleolin from the nucleolus into HSV-1 replication compartments in a ICP4-dependent manner, thereby leading to enhanced HSV-1 replication and disruption of the nucleolar structure [42]. Similarly, nucleolar upstream binding factor (UBF) and nucleophosmin (B23.1) are recruited to adenovirus replication compartments to promote viral DNA replication [43–45]. The autonomous parvovirus minute virus of mice has been shown to replicate its DNA in the nucleoli of mouse fibroblasts [46,47] Additionally, borna disease virus transcription and replication take place in the

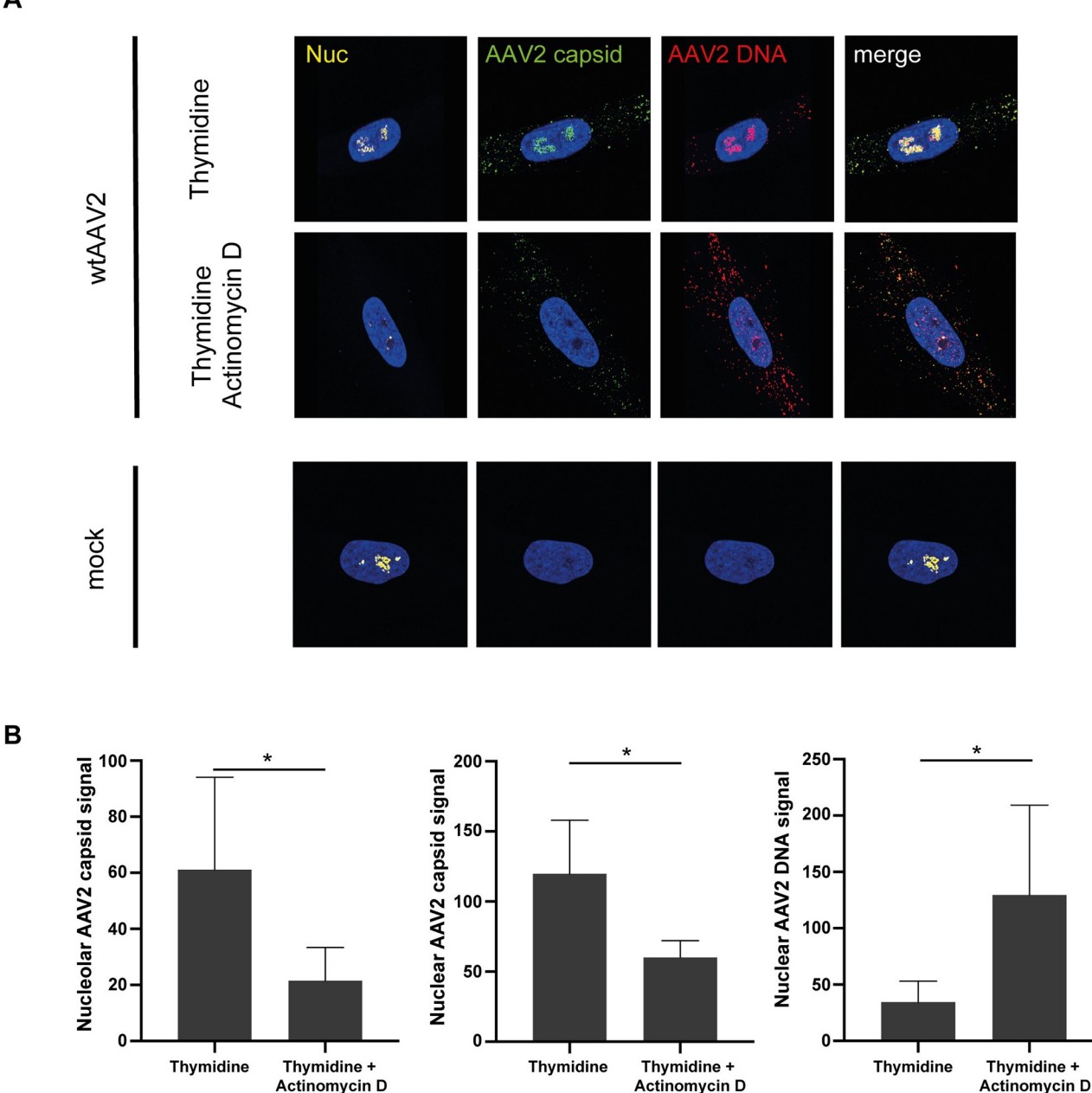

**Fig 9. Actinomycin D treatment overcomes thymidine-mediated obstruction of AAV2 uncoating.** NHF cells were arrested in G1 cell cycle phase by a double thymidine block before and during infection with wtAAV2 (MOI 20'000). At 24 hpi, the cells were treated with actinomycin D for 1 h, fixed and processed for multicolor IF analysis combined with FISH and CLSM. (A) Nucleoli (Nuc) were visualized using an antibody against fibrillarin (yellow). Intact capsids were stained using an antibody that detects a conformational capsid epitope (green). AAV2 DNA (red) was detected with an Alexa Fluor (AF) 647 labeled, amine-modified DNA probe that binds to the AAV2 genome. (B) Image-based quantification of 50 individual cells per condition for nucleolar capsid, total nuclear capsid or total nuclear AAV2 genome signals. p-values were calculated using an unpaired Student's t-test (*—$p \leq 0.05$, **—$p \leq 0.01$, ***—$p \leq 0.001$, ****—$p \leq 0.0001$).

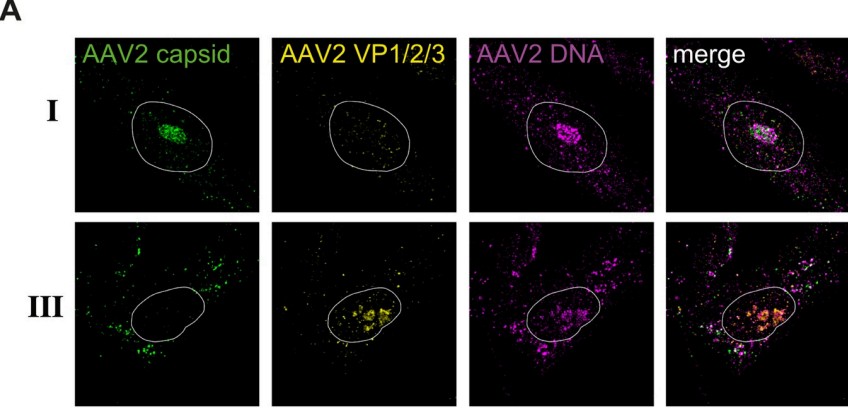

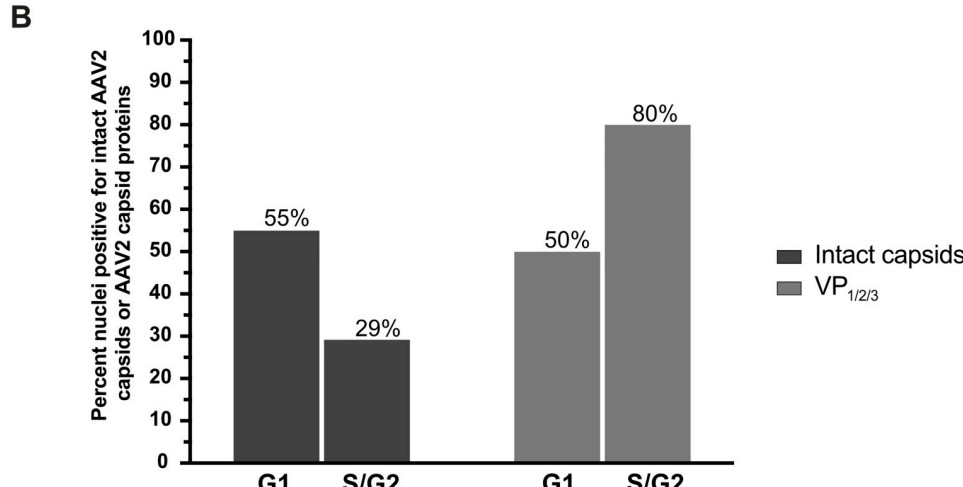

**Fig 10. Capsid disassembly coincides with cell cycle progression.** NHF cells were infected with wtAAV2 (MOI 20'000). At 24 hpi, the cells were fixed and processed for multicolor IF analysis combined with FISH and CLSM. (A) Intact capsids (green) or capsid proteins (yellow) were detected using either an antibody against intact AAV2 capsids (conformational capsid epitope) or an antibody (linear epitope) against VP1, VP2 and VP3. AAV2 DNA (magenta) was detected with an Alexa Fluor (AF) 647 labeled, amine-modified DNA probe that binds to the AAV2 genome. Nuclei were counterstained with DAPI and illustrated as white lines. The cells shown were assigned as pattern I and III according to the intensity of the capsid staining and following a similar classification as defined in Fig 4. (B) Quantification of at least 70 nuclei positive for intact AAV2 capsids or capsid proteins during cell cycle progression.

nucleoli as well [48]. Moreover, specific mRNAs and proteins of many different viruses, including HIV-1, Japanese encephalitis virus, and Semliki Forest virus traffic through the nucleolus for processing, and the inhibition of such trafficking affects virus replication [49–51].

Helper virus-supported AAV2 DNA replication occurs in nuclear replication compartments that are distinctly separate from nucleoli (S14 Fig). However, AAV2 interacts with nucleoli at both early and late stages of the replication cycle, cell entry and assembly. Upon nuclear entry AAV2 capsids have been shown to accumulate in the nucleoli [14]. Later in infection, intact AAV2 capsids were detected also in the nucleoplasm. Treatment of cells with hydroxyurea or proteasome inhibitors, both of which are known to improve AAV2

transduction efficiency, increased either nucleolar accumulation of AAV2 capsids or their relocalization into the nucleoplasm. Moreover, the post-transcriptional silencing of nucleophosmin enhanced nucleolar accumulation and increased transduction comparable to the proteasome inhibitor treatment, while the siRNA-mediated silencing of nucleolin mobilized capsids to the nucleoplasm and enhanced transduction similar to the treatment with hydroxyurea. These observations led to the hypothesis that AAV2 uncoating takes place in the nucleoplasm [14]. However, in the aforementioned study and all other studies, viral capsids and viral genomes were not directly correlated on the single cell level but rather analyzed by quantitative (q)PCR, Western blot and immunofluorescence [14,24–30].

By employing combined immunofluorescence analysis with fluorescence *in situ* hybridization (IF-FISH) and confocal laser scanning microscopy, we monitored the spatial and temporal distribution of AAV2 capsids and genomes on the single cell level and observed that AAV2 DNA accumulates together with AAV2 capsids in the nucleoli of AAV2 infected cells, thereby confirming previous findings that the nucleolus acts as a sink for incoming AAV2 particles. In addition, our IF-FISH assay provides evidence for the stepwise uncoating of the AAV2 particle. Step 1 occurs in the cytoplasm, probably within endocytic vesicles of this compartment, and leads to AAV2 genome accessibility where the viral capsid is still recognized by an antibody that binds to a conformational capsid epitope and co-localizes with AAV2 DNA. Step 2 takes place within the nucleoli and results in the complete disassembly of the AAV2 capsids and the accumulation of AAV2 DNA and AAV2 capsid proteins.

The exact mechanism that drives step 1 of the AAV2 uncoating process and how it fits into current AAV2 trafficking models remains to be investigated. However, our data show that it is enhanced by acidification, as co-detection of AAV2 capsids and AAV2 DNA in the cytoplasm was reduced in cells treated with bafilomycin A1, a vacuolar H+-ATPase inhibitor. The interaction of importin β and the N-terminal end of VP1 [24] as well as the pH-dependent structural reorganization of the AAV2 capsid leading to the extrusion of the nuclear localization signals located in VP1$_u$ and VP1/VP2 N-termini [13,52,53] have been shown to be relevant for efficient nuclear entry of the AAV2 capsid. Whether or not the accessibility of the AAV2 genome for the AAV2 DNA specific FISH probe in AAV2 capsids that are still recognized by a conformational capsid antibody is due to pH-dependent structural rearrangements of the capsid or rather due to the protrusion of the AAV2 DNA from an almost intact AAV2 capsid, as it has been shown for thermally induced AAV2 genome release [54], requires further investigation. Similarly, whether or not AAV2 particles with FISH-probe accessible genomes can indeed enter the nucleus and establish viral replication compartment also requires further investigation. The accessibility of the AAV2 genome, however, might provide further evidence for the Toll-like receptor 9 (TLR9) mediated antiviral activation state in AAV2 infected untransformed cells [55].

Our image-based analysis of the nucleolar structure as well as AAV2 DNA, AAV2 capsids, and AAV2 capsid proteins, relative to the cell cycle profile provides strong evidence that step 2 of the uncoating process, the complete disassembly of the capsid, occurs in the nucleolus. Most interestingly, we provide also evidence that AAV2 second-strand synthesis may occur in the nucleoli, as we identified replication form monomers, the products of second-strand synthesis, in nucleolar fractions as early as 24 hpi, while no such products were identified in the nucleoplasmic fraction by 48 hpi. In principle, double-stranded AAV2 DNA may also be generated by the annealing of DNA strands with opposite orientation. However, this could be expected to occur also in the nucleoplasmic fraction, where it was not observed by 48 hpi, although the intensity of the single-stranded DNA band was comparable to that observed in the nucleolar fraction. The finding that rAAV DNA isolated from nucleoli is transcritptionally active further supports the hypothesis that second-strand synthesis can indeed take place in nucleoli.

Nevertheless, we cannot exclude the possibility that AAV2 DNA uncoating and genome processing can occur also in other nuclear compartments.

The data also support the hypothesis that the complete disassembly of AAV2 capsids is induced by the structural reorganization of the nucleolus in a cell cycle-dependent manner. While it is common for viruses to take advantage of the cell cycle or to undermine it in order to drive different stages of their life cycle (reviewed in [56]), little is known about viruses availing the cell cycle to drive their uncoating process. A recent study demonstrated that the HIV-1 is unable to uncoat its core in quiescent CD4+ lymphocytes and that the uncoating activity requires transition from $G0/G1_a$ to $G1_b$ stage, arguing for the demand of cell cycle-dependent specific factors for HIV-1 uncoating [57]. For foamy virus (FV), capsid uncoating and the formation of the preintegration complex starts with the onset of mitosis. As the microtubule organizing center and the associated centrosomes, both being relevant for the life cycle of the virus, are highly linked to cell cycle regulation, it is likely that cell cycle regulatory proteins might contribute to FV capsid uncoating [58].

Nucleolar proteins such as nucleolin can bind to AAV2 capsids and seem to play a major role in the AAV2 replication cycle. Several studies demonstrated that nucleolin is barely detectable in resting cells; in contrast, nucleolin represents the major nucleolar protein in cycling eukaryotic cells [23,59]. This observation provides evidence for a link between AAV2 capsids, cell cycle progression and nucleolar proteins.

The interaction of some virus proteins with the nucleoli has been shown to be regulated by the cell cycle as well. For example, the human cytomegalovirus UL83 protein and the coronavirus nucleocapsid protein have been shown to localize to the nucleolus preferentially in the G1 and the G2 phase of the cell cycle, respectively. Most interestingly, we have previously reported that AAV2 gene expression and DNA replication occur primarily in the G2 phase of the cell cycle [60]. This cell cycle-dependence was not due to inefficient second-strand synthesis in cells in G1 nor to cell cycle-dependent DNA damage responses, as gene expression from a double-stranded self-complementary AAV2 vector was also reliant on cells in the G2 phase of the cell cycle and the inhibition of specific kinases in DNA damage signaling did not result in a shift of gene expression to cells in G1.

Based on our new finding that the accumulation of AAV2 DNA together with disassembled AAV2 capsid proteins in dispersed nucleoli coincides with the G2 phase of the cell cycle, it is tempting to speculate that cell cycle-dependent AAV2 gene expression and DNA replication is controlled by cell cycle-dependent reorganization of the nucleolar structure that enables AAV2 uncoating. This hypothesis is further supported by the observation that perturbations that lead to changes in the nucleolar architecture such as actinomycin D treatment (this study), helper virus infection, or post-transcriptional silencing of nucleolin, enhance AAV2 transduction. However, the exact mechanism of the disassembly of the AAV2 capsid by nucleolar reorganization during cell cycle progression remains to be further investigated.

AAV infects proliferating and non-dividing or quiescent cells, and this feature is considered an advantage in gene transfer and gene therapy approaches employing AAV vectors [61]. Particularly, long-term stable transgene expression mediating therapeutic efficacy in human clinical trials such as those focusing on liver [62]. Nevertheless, when comparing efficacy, transduction efficiency was shown to be lower in quiescent cells compared to proliferating cells [61] an observation that our here reported findings might help to explain. Besides, our data do not exclude the possibility that uncoating may take place in the nucleoplasm by mechanisms that do not involve the cell cycle.

We here focused our study on wild-type AAV2 and showed compelling evidence that the same steps towards uncoating are also employed by recombinant AAV2 vectors. Albeit it remains to be proven experimentally, but given the similarities between various serotypes

regarding the main steps of cell transduction (despite differences in receptor usage, capsid stability and efficacy of episome formation), we assume that also serotypes other than AAV2 undergo a partial uncoating (evidence by the reported TLR9 mediated sensing of vector genomes [63] in the endosomal compartment as well as structural biology studies reported by [64,65]) that is completed in the nucleolus and supported by alterations of the nucleolar structures.

## Materials and methods

### Cells and viruses

Normal human fibroblast (NHF) cells were kindly provided by X.O. Breakefield (Massachusetts General Hospital, Charlestown, MA, USA). NHF cells, neonatal human dermal fibroblast cell line HDFn (ATCC PCS-201-010, American Type Culture Collection, Rockville, Md, USA), human hepatocellular carcinoma cell line Hep G2 (ATCC HB-8065, American Type Culture Collection, Rockville, Md, USA), lung epithelial cells A549 (ATCC CCL-185, American Type Culture Collection, Rockville, Md, USA) and African green monkey kidney cells (Vero cells, ATCC, American Type Culture Collection, Rockville, Md, USA) were maintained in growth medium containing Dulbecco's modified Eagle medium (DMEM) supplemented with 10% fetal bovine serum (FBS), 100 U/ml penicillin G, 100 μg/ml streptomycin, and 0.25 μg/ml amphotericin B (1% AB) at 37˚C in a 95% air-5% $CO_2$ atmosphere. Wild-type (wt) AAV2 was produced by H. Buening (University of Hannover, Hannover, Germany) and the Viral Vector Facility (VVF) of the Neuroscience Center Zurich (ZNZ). The recombinant vector of AAV serotype 2 (rAAVGFP) was produced by the VVF of the ZNZ. Besides, rAAVCF-PRep (serotype 2) was produced by transient transfection of 293T cells with pDG [66] and pAAVCFPRep [60], and purified by an iodixanol density gradient. Titers of genome-containing particles were determined as described previously [67].

The VP1 AAV2 mutant ([76]HD/AN) was constructed according to Girod et al. [32] and produced by the VVF. Briefly, the [76]HD/AN mutant construct was generated by mutating two key residues [76]HD to [76]AN using K-[76]HD/AN (5'GCGGCCCTCGAGGCCAACAAAGCCTAC GACCGG 3'), L-[76]HD/AN (5'CCGGTCGTAGGCTTTGTTGGCCTCGAGGGCCGC 3'), p*sub*-201 [68] containing the full-length AAV2 genome as template and the QuikChange Site-Directed Mutagenesis Kit (Agilent Technologies). HSV-1 delta ICP27 (HSV-1ΔICP27) mutants were provided from R. Everett (University of Glasgow).

### Antibodies

The following primary antibodies were used: anti-AAV2 intact particle (A20, ProGen; dilution for Immunofluorescence [IF] 1:50), anti-AAV VP1/VP2/VP3 (VP51, ProGen, dilution for IF 1:10), anti-AAV VP1/VP2 (A69, ProGen, dilution for IF 1:10), anti-AAV2 Rep (Fitzgerald Industries, 10R-A111A, dilution for IF 1:10), anti-fibrillarin (Abcam ab5821; dilution for IF 1:200; dilution for Western blotting [WB] 1:650), anti-cyclin A (Santa Cruz sc-751, dilution for IF 1:500), anti-α-tubulin (Sigma-Aldrich, T5168, dilution for WB 1:1'000), anti-NPM (Abcam ab10530, dilution for WB 1:2000), anti-NCL (Abcam ab22758, dilution for WB 1:1000), anti-vimentin (Santa Cruz sc-5565, dilution for WB 1:400) The following secondary antibodies were used: Alexa Fluor 594 goat anti-rabbit IgG (Life Technologies A11037, dilution for IF 1:500), Alexa Fluor 488 goat anti-mouse IgG (Invitrogen A11001, dilution for IF 1:500), Goat anti-mouse IgG (H+L) 680RD: LI-COR (926–68070), dilution for WB 1:10'000, Goat anti-rabbit IgG (H+L) 680RD: LI-COR (926–68071), dilution for WB 1:10'000.

## Viral infection and treatments

NHF, HDFn, Hep G2, A549 or Vero cells were seeded onto coverslips (12-mm diameter; Glaswarenfabrik Karl Hecht GmbH & Co. KG, Sondheim, Germany) in 24-well tissue culture plates at a density of $3x10^4$ cells per well. Cycling or contact inhibited NHF cells were seeded at a density of $2x10^4$ or $1x10^5$, respectively. The next day, the cells were washed with PBS and either mock-infected, infected with either wtAAV2 at a multiplicity of infection (MOI) of 2'000 or 20'000 genome containing particles (gcp) per cell or rAAVGFP (MOI of 20'000) in 250 µl of DMEM (0% FBS, 1% AB) pre-cooled to 4°C. The plates were first incubated for 30 min at 4°C to synchronize viral uptake and then incubated at 37°C in a humidified 95% air-5% $CO_2$ incubator for the indicated time period. For acidification experiments NHF cells were treated with bafilomycine A1 (50 or 200 nM) or DMSO in DMEM (10% FBS, 1% AB) 1 h prior to infection with wtAAV2. G1 cell cycle phase arrest prior to infection was induced by a double thymidine block. For this, cells were seeded in 10 cm tissue culture dishes ($5x10^5$ cells per dish) and 12 h later the growth medium was replaced with medium (DMEM, 10% FBS, 1% AB) containing 3 mM thymidine. After 12 h of incubation, the cells were washed with PBS, trypsinized and split at a density of $6x10^4$ cells per well into 6-well tissue culture plates containing coverslips. In order to complete the double thymidine block, the growth medium was replaced 12 hours later by medium containing 3 mM thymidine. After 12 hours, the cells were either released from the block by washing out the thymidine with PBS or directly infected with wtAAV2 in the presence of thymidine. Nucleolar disruption was induced with actinomycin D (50 nM) in DMEM (2% FBS, 1% AB) for 1 h after 24 h of infection in presence of thymidine.

## Cell cycle analysis based on 4',6-diamidino-2-phenylindole (DAPI) staining

The workflow described is closely related and adapted from the protocol published by Roukos et al., 2015 [36]. Briefly, NHF cells were seeded onto coverslips (12-mm diameter; Glaswarenfabrik Karl Hecht GmbH & Co. KG, Sondheim, Germany) in 24-well tissue culture plates ($3x10^4$ cells per well). The next day, the cells were washed with PBS, processed as indicated in the results and the figure legends, counterstained with DAPI and imaged by confocal laser scanning microscopy (Leica SP8; Leica Microsystems, Wetzlar, Germany). An automated CellProfiler (V.2.2.0-V.4.0.7) pipeline measured the integrated intensity of DAPI. Next, the histograms of DAPI, corresponding to the DNA content, were plotted and visual thresholds for each cell cycle phase were selected. These thresholds were finally read back into a secondary CellProfiler pipeline, which lastly allowed tracking of individual cells and measurements.

## Comparison of cell cycle classification using flow cytometry and the DAPI integrated intensity protocol

To validate the DAPI integrated intensity protocol, NHF cells were synchronized using a double thymidine block (as described above). After the release, the cells were either mock-treated or treated with nocodazole (200 nM) for 24 hours. For flow cytometry, the cells were harvested by exposing them to 0.05% Trypsin-EDTA solution for 10 min, centrifuged and washed with PBS, fixed in 2.5 ml ice-cold 100% ethanol, centrifuged, washed once again with PBS and stained with a freshly made solution containing 1 µg/mL DAPI, 0.05% Triton X-100 and 0.1 mg/mL ribonuclease A (RNase A) in PBS. All samples were incubated for 45 min at 37°C in the dark. After incubation, the cells were washed twice with PBS and then resuspended in 200 µl PBS prior to analysis (SONY SP6800 Spectral Analyzer). For confocal laser scanning microscopy, the coverslips were embedded in ProLong Anti-Fade mountant with DAPI

(Molecular Probes, Eugene, OR, USA) and imaged using a 63x oil immersion objective (Leica SP8; Leica Microsystems, Wetzlar, Germany). Images were analyzed as described in the section cell cycle analysis based on 4',6-diamidino-2-phenylindole (DAPI) staining.

## Combined multicolor immunofluorescence analysis and fluorescence *in situ* hybridization (FISH)

FISH was performed essentially as described previously by Lux et al. [27]. Briefly, a 3.9-kb DNA fragment containing either the wtAAV2 genome or the rAAVGFP genome without the inverted terminal repeats was amplified by PCR from plasmid pDG using forward (5'-CGGGGTTTTACGAGATTGTG-3') and reverse (5'-GGCTCTGAATACACGCCATT-3') primers or from pAAVGFP (provided by M. Linden, King's College London School of Medicine, London, UK) using forward (5'- ATGGTGAGCAAGGGCGAGGA-3') and reverse (5'-CTTGTACAGCTCGTCCATGC-3') primers and the following conditions: 30 s at 95˚C; 35 cycles of 10 s at 98˚C, 15 s at 58˚C, and 75 s at 72˚C; and 10 min at 72˚C. The PCR sample was then digested with DpnI to cut the residual template DNA and purified with the Pure Link PCR purification kit (Qiagen, Hilden, Germany). The DNA fragment was labeled with 5-(3-aminoallyl)dUTP by nick translation according to the manufacturer's protocol (Ares DNA labeling kit, Molecular Probes, Eugene, OR, USA), and the incorporated dUTPs were labeled with amino-reactive Alexa Fluor 647 dye by using the same Ares DNA labeling kit. NHF cells were plated onto glass coverslips in 24-well tissue culture plates at a density of $3x10^4$ cells per well and 24 h later mock-infected or infected with wtAAV2 (MOI of 20'000). 24 hours after infection, the cells were washed with PBS, fixed for 30 min at room temperature (RT) with 2% PFA (in PBS), and washed again with PBS. The cells were then quenched for 10 min with 50 mM $NH_4Cl$ (in PBS), washed with PBS, permeabilized for 10 min with 0.2% Triton X-100 (in PBS), blocked for 10 min with 0.2% gelatin (in PBS) followed by two washing steps with PBS before blocking for 30 min in PBST (0.05% Tween 20 in PBS) supplemented with 3% BSA at 4˚C. After antibody staining in PBST-BSA (3%, 25 μl/coverslip) for 1h at RT in the dark in a humidified chamber, the cells were washed three times for 5 min with PBST (0.1%), post-fixed with 2% PFA and blocked with 50 mM glycine in PBS for 5 min at RT.

Hybridization solution (20 μl per coverslip) containing 1 ng/ml of the labeled DNA probe, 50% formamide, 7.3% (w/v) dextran sulfate, 15 ng/ml salmon sperm DNA, and 0.74x SSC (1x SSC is 0.15 M NaCl and 0.015 M sodium citrate) was denatured for 3 min at 95˚C and shock-cooled on ice. The coverslips with the fixed and permeabilized cells facing down were placed onto a drop (20 ml) of the denatured hybridization solution and incubated overnight at 37˚C in a humidified chamber (note that the cells were not denatured, as the AAV2 genome is present as ssDNA). The next day, the coverslips were washed three times with 2x SSC at 37˚C, three times with 0.1x SSC at 60˚C, and twice with PBS at RT. To confirm the FISH signal, some samples (as stated in the results) were treated with DNase I (1U/ μl) for 1 h at 37˚C followed by inactivation in 30% formamide, 0,1% Triton-X 100 and 2x SSC for 10 min at RT.

The cells were then embedded in ProLong Anti-Fade mountant with or without DAPI (Molecular Probes, Eugene, OR, USA) and imaged as midsections if not stated otherwise by confocal laser scanning microscopy (Leica SP8; Leica Microsystems, Wetzlar, Germany). To prevent cross talk between the channels for the different fluorochromes, all channels were recorded separately, and fluorochromes with longer wavelengths were recorded first. The resulting images were processed using Imaris V.7.7.2-V.9.6.0 (Bitplane, Oxford Instruments, Biplane AG, Zurich, Switzerland).

## Isolation of nucleoli from AAV2 infected cells

NHF cells were seeded in T150 cell culture flasks at a density of $3x10^6$ cells per flask. The next day, the cells were washed with PBS and either mock-infected or infected with wtAAV2 or rAAVCFPRep at a MOI of 20'000 gcp per cell in 5 ml of DMEM (0% FBS, 1% AB) pre-cooled to 4˚C. The flasks were first incubated for 30 min at 4˚C to synchronize viral uptake and then incubated at 37˚C in a humidified 95% air-5% $CO_2$ incubator for the indicated time period.

Isolation of nucleoli was performed according to Hacot et al [69]. Briefly, cells were washed with ice-cold PBS and detached using Trypsin-EDTA (0.05%) at the indicated time points. The cells were pelleted, and the volume was visually determined. The cell pellet was resuspended in 2 volumes of nucleolar standard buffer (NSB, 10 mM Tris-HCl, 10 mM NaCl, 1 mM MgCl2) and the reference volume (RV) was calculated (measured volume–volume NSB added). After adjusting the volume to 15 times the RV with NSB, the cell suspension was incubated on ice for 30 min. The cells were lysed by adding 10% NP-40 to a final concentration of 0.3% and then centrifuged at 1200 g for 5 min at 4˚C. The supernatant containing the cytoplasmic fraction was collected. To purify the nuclei, the pellet was resuspended in 10 RV of 250 mM sucrose containing 10 mM MgCl2. After carefully adding 10 RV of 880 mM sucrose solution containing 5 mM MgCl2, the suspension was centrifuged for 10 min at 4˚C and 1200 g. The purified nuclei were resuspended in 10 RV of 340 mM sucrose solution containing 5 mM MgCl2 and sonicated (20% amplitude, QSonica, Q800R3, CT, USA) at 4˚C for three pulses of 30 s in 30 s intervals to break the nuclei. 10 RV of 880 mM sucrose solution was added on the bottom of the tube containing the sonicated nuclei, and the suspension was centrifuged for 20 min at 4˚C and 2000 g. The supernatant, containing the nucleoplasmic fraction was collected, and the pelleted nucleoli were resuspended in 50 μl TBS (50 mM Tris-HCl, 150 mM NaCl, pH 7.5). Cell fractionation was controlled by Western blot and IF-FISH analyses.

## Isolation of fully uncoated DNA

After freeze thawing the isolated nucleoli three times, 1 vol AMPure XP beads (Beckman&-Coulter, A63880, IN, USA) were added, and the samples were incubated for 5 minutes at RT while rotating the tube. The beads were collected using a magnetic rack, and the supernatant was set aside (containing unopened capsids and nucleolar debris). The beads were washed twice with freshly prepared 70% ethanol (without removing the tube from the magnetic rack) and left to dry for approx. 30 s (but not to the point of cracking). The DNA was eluted in DNase free water for 2 min at RT. The beads were collected using a magnetic rack and then discarded. The supernatant containing the DNA was further purified by phenol/chloroform extraction and ethanol precipitation as described below.

## Hirt DNA extraction

Extraction of extrachromosomal DNA was performed according to the Hirt protocol [70]. In short, cells were washed with PBS and detached using Trypsin-EDTA (0.05%). The cells were pelleted and resuspended in 50 μl TBS (50 mM Tris-HCl, 150 mM NaCl, pH 7.5). After adding 500 μl Hirt Buffer (0.6%SDS, 10mMTris-HCl, 10mMEDTA, pH7.5) the cell suspension was incubated for 1h at room temperature. After adding 120 μl 5 M NaCl solution, the sample was incubated for at least 12 h at 4˚C. For phenol/chloroform extraction of the DNA, the sample was centrifuged for 10 min at 4˚C and 15'500 g. The supernatant was transferred to a fresh tube, and 1 volume of phenol:chloroform:isoamylalcohol (25,24:1, v/v, 15593031, Invitrogen, USA) was added. The sample was centrifuged for 5 min at 4˚C and 15'500 g. The supernatant was transferred into a fresh tube, and 1 volume chloroform was added. The sample was centrifuged for 1 min at 4˚C and 15'500 g, and the supernatant was transferred into a fresh tube. 2.5

volumes of EtOH (pure) and 0.1 volume of 3 M NaAc pH 5.5 were added to the sample. To precipitate the DNA the suspension was incubated for at least 20 min at -80˚C. The sample was centrifuged for 10 min at 4˚C and 18'000 g, and the supernatant was discarded. The DNA pellet was washed with 70% EtOH and centrifuged for 10 min at 4˚ and 18'000 g. The supernatant was removed, and the DNA pellet was left to dry for at least 20 min at room temperature. After drying, the pellet was resuspended in Tris-HCl pH 8.5 and incubated for 10 min at 37˚C.

## Southern analysis

For Southern analysis extrachromosomal DNA was extracted using the Hirt protocol as described above. Prior to Southern blotting the samples were denatured for 5 min at 95˚C, shock-cooled on ice, and then separated on a 0.8% agarose gel (100 V, 3.5h) and transferred onto nylon membranes (Hybond-N+, RPN119B, Amersham, Little Chalfont, UK). A DIG-labeled marker was used as reference (DNA molecular weight marker III, 11218602910, Roche). AAV2 sequences were detected using a DIG-labeled AAV2 *rep*-specific probe, which was visualized using an anti-DIG antibody conjugated with alkaline phosphatase and activation with the chemiluminescence substrate CDP Star (Roche) according to the manufacturer's protocol. The DIG-labeled probe was synthesized using the PCR DIG probe synthesis kit (11636090910, Roche, Switzerland) and following primers: 5'-gaacgcgatatcgcagccgccatgccggg-3' and 5'-ggatccgaattcactgcttctccgaggtaatc-3'. For chemiluminescence visualization the LI-COR imaging system Odyssey FC (LI-COR Biosciences, Lincoln, NE, USA) was used.

## Transfection of nucleolar AAV2 DNA

NHF cells were plated onto glass coverslips in 24-well tissue culture plates at a density of $3 \times 10^4$ cells per well and 24 h later transfected with nucleolar uncoated (bead-purified) rAAVCFPRep DNA by using Lipofectamine LTX according to the manufacturer's instructions (Life Technologies). At 24 hours after transfection, the cells were mock-infected or infected with HSV-1ΔICP27. At 48 hpi the cells were monitored by epifluorescence microscopy.

## Negative contrast stain

For the examination of AAV2 capsid disintegration a negative contrast staining was performed. For this, 10 μl of the wtAAV2 stock were placed onto a parafilm strip and adsorbed to carbon coated parlodion films mounted on 300 mesh/inch copper grids by placing upside-down on the drop and incubated for 10 min at RT. Washing was done by transferring the grid to a $H_2O$ drop. Subsequently the grid was placed onto a drop of phosphotungstic acid (PTA, pH 7.0) for 60 seconds. Remaining liquid was removed by tipping the edge of the grid on a filter paper. The samples were analyzed in a Philips CM 12 transmission electron microscope (Eindhoven, the Netherlands) equipped with a charge-coupled device (CCD) camera (Ultrascan 1000, Gatan, Pleasanton, CA, USA) at an acceleration voltage of 100 kV.

## Image-based quantification and data analysis

For image-based quantification and data analysis, at least 50 individual cells per sample or condition were recorded and analyzed using different CellProfiler (V.2.2.0-V.4.0.7) pipelines. The output csv-files were further analyzed using Matlab (R2017a) and GraphPad Prism 6 to 9. Depending on distribution frequency and standard deviation (SD), statistical analysis of individual cells was either performed by unpaired Student's t-test or an unpaired t-test with

Welch's correction (not assuming equal SDs). If not stated otherwise, each graph illustrates one representative experiment.

## Supporting information

**S1 Fig. Nucleolar accumulation of AAV2 capsids and genomes at different MOIs.** (A) NHF or (B) A549 cells were either mock-infected of infected with wtAAV2 at a MOI of 20'000 or 2'000. At 24 hpi, the samples were processed for IF-FISH and CLSM. Intact capsids were stained using an antibody that detects a conformational capsid epitope (green). Nucleoli (Nuc) were visualized using an antibody against fibrillarin (yellow). AAV2 DNA (red) was detected with an Alexa Fluor (AF) 647 labeled, amine-modified DNA probe that binds to the AAV2 genome. Nuclei were counterstained with DAPI (blue).
(TIF)

**S2 Fig. Transduction efficiency correlates with uncoating efficiency.** NHF, A549 or Hep G2 cells were either mock-infected or infected with a recombinant AAV2 vector (rAAVGFP) at a MOI of 20'000. At 24 hpi, the samples were processed for IF-FISH and CLSM. (A) Intact capsids were stained using an antibody that detects a conformational capsid epitope (green). Nucleoli (Nuc) were visualized using an antibody against fibrillarin (yellow). AAV2 DNA (red) was detected with an Alexa Fluor (AF) 647 labeled, amine-modified DNA probe that binds to the AAV2 genome. Nuclei were counterstained with DAPI (blue). (B) Comparison of transduction efficiency (% GFP positive cells) and uncoating efficiency (% rAAV capsid-negative DNA-positive nucleoli) of at least 50 cells.
(TIFF)

**S3 Fig. Endosomal escape is relevant for nucleolar accumulation.** NHF cells were either mock-infected or infected with wtAAV2 or a VP1 AAV2 mutant ([76]HD/AN) at a MOI of 20'000. At 5 hpi, the samples were processed for IF-FISH and CLSM. Intact capsids were stained using an antibody that detects a conformational capsid epitope (green). AAV2 DNA (red) was detected with an Alexa Fluor (AF) 647 labeled, amine-modified DNA probe that binds to the AAV2 genome. Nuclei were counterstained with DAPI (blue) or illustrated as white lines.
(TIF)

**S4 Fig. DNase I treatment eliminates the AAV2 genome signal in IF-FISH assays.** (A) NHF or (B) A549 cells were infected with wtAAV2 (MOI 20'000). At 24 hpi, the cells were either treated with DNase I (1 U/μl) for 1 h at 37°C, RNase A (0.5 mg/ml) for 1 h at 37°C, or a combination of both. DNase or RNase inactivation was achieved by washing the cells twice for at least 10 min with either DNase inactivation buffer (30% Formamide, 0.1% Triton X-100, 2X SSC) or RNase inactivation buffer (0.5X SSC, 0.1 SDS), respectively. Afterwards the samples were fixed and processed for multicolor IF analysis combined with FISH and CLSM. Intact capsids were stained using an antibody that detects a conformational capsid epitope (green). Nucleoli (Nuc) were visualized using an antibody against fibrillarin (yellow). AAV2 DNA (red) was detected with an Alexa Fluor (AF) 647 labeled, amine-modified DNA probe that binds to the AAV2 genome. Nuclei were counterstained with DAPI.
(TIF)

**S5 Fig. Verification of cell fractionation by IF-FISH and Western blot.** NHF cells were either mock-infected or infected with wtAAV2 at an MOI of 20'000. At 24 hpi, the cells were fractionated and isolated nuclei (A) or nucleoli (B) were applied to fibronectin coated coverslips and processed for IF-FISH and CLSM for morphological assessment of the two fractions. Intact capsids

were stained using an antibody that detects a conformational capsid epitope (green). Nucleoli (Nuc) were visualized using an antibody against fibrillarin (yellow). AAV2 DNA (red) was detected with an Alexa Fluor (AF) 647 labeled, amine-modified DNA probe that binds to the AAV2 genome. Nuclei were counterstained with DAPI (blue). (C) Western analysis revealed the absence of α-tubulin (cytoplasmic marker) and presence of fibrillarin (nuclear and nucleolar marker) in the nuclear and nucleolar fractions ($10^5$ cell equivalents per lane were loaded). (TIF)

**S6 Fig. Functionality of nucleolar AAV2 DNA.** NHF cells were infected with rAAVCFPRep (MOI 20'000), and nucleolar fractions were prepared 24 h later. Bead-purified (uncoated) nucleolar DNA was transfected into NHF cells and, after 24 h, the cells were mock-infected or infected with HSV-1ΔICP27. At 48 hpi, the cultures were monitored for CFP-fluorescence using an epifluorescence microscope. (TIFF)

**S7 Fig. Co-detection of AAV2 DNA with AAV2 capsids and AAV2 capsid proteins in Vero cells.** Vero cells were mock-infected or infected with wtAAV2 (MOI 20'000; two individual cells are shown). At 24 hpi, the cells were fixed and processed for multicolor IF analysis combined with FISH and CLSM. Intact capsids (green) or capsid proteins (yellow) were detected using either an antibody against intact AAV2 capsids (conformational capsid epitope) or an antibody (linear epitope) against VP1, VP2 and VP3. AAV2 DNA (red) was detected with an Alexa Fluor (AF) 647 labeled, amine-modified DNA probe that binds to the AAV2 genome. Nuclei were counterstained with DAPI (blue). (TIF)

**S8 Fig. AAV2 infection does not alter the ratio of dense to dispersed nucleoli.** NHF cells were mock-infected or infected with wtAAV2 (MOI 20'000) and 24 h later processed for combined IF-FISH, CLSM and quantification of the nucleolar structure of 100 individual nuclei in mock- or AAV2 infected cells. p-values were calculated using an unpaired Student's t-test (*—$p \leq 0.05$, **—$p \leq 0.01$, ***—$p \leq 0.001$, ****—$p \leq 0.0001$). (TIF)

**S9 Fig. Schematic representation of the cell cycle staging by cyclin A staining and the integrated intensity of DAPI.** (1) The background of each image was subtracted. (2) Nuclei and cyclin A stainings were identified as primary objects in CellProfiler. (3) The stainings were related to each other and (4) the DAPI integrated intensity of each nucleus was measured. (5) Histograms of the DAPI integrated intensities were plotted using an automated script in Matlab and visual thresholds were set. (6) Cells were classified in CellProfiler using the visual thresholds obtained in step 5. (7) The classification of each nucleus into G1, S or G2 was overlayed on the original DAPI image to track individual cells. (TIF)

**S10 Fig. Cell cycle staging of DAPI stained cells using flow cytometry analysis and confocal laser scanning microscopy.** NHF cells were synchronized using a double thymidine (3 mM) block. After the release, the cells were either mock-treated or treated with nocodazole (200 nM) for 24 hours to induce a G2 cell cycle phase arrest. Flow cytometry shows the mean value of three experiments, each replicate contains at least 5'000 scored events. CLSM analysis shows the mean value of three experiments, each replicate contains at least 100 individual analyzed cells. p-values were calculated using an unpaired Student's t-test (*—$p \leq 0.05$, **—$p \leq 0.01$, ***—$p \leq 0.001$, ****—$p \leq 0.0001$). (TIFF)

**S11 Fig. Nucleolar reorganization during cell cycle progression.** Image-based analysis of the ratios of dense to dispersed nucleoli of 150 individual cells in different cell cycle phases (G1, S and G2). Statistical analysis was performed on three independent experiments and p-values were calculated using an unpaired Student's t-test (*—$p \leq 0.05$, **—$p \leq 0.01$, ***—$p \leq 0.001$, ****—$p \leq 0.0001$).
(TIF)

**S12 Fig. AAV2 uncoating is inhibited in contact inhibited NHF cells.** NHF cells were seeded at 40% or 100% confluency, respectively. The cells were either mock-infected or infected with wtAAV2 (MOI 20'000) and, at various time points post infection, processed for combined IF-FISH and CLSM. (A) Image-based analysis of the total uncoating rate in nucleoli of 30 cells at 48 hpi at 40% and 100% confluency. Quantification of cells (30 cells per time point) with (B) AAV2 capsid-positive and DNA-positive (AAV2 capsid+DNA+) signal or (C) AAV2 capsid protein VP1, VP2 and VP3-positive and DNA-positive (AAV2 $VP_{1/2/3}$+DNA+) signal in nucleoli.
(TIFF)

**S13 Fig. Capsid disassembly coincides with cell cycle progression in HDFn cells.** HDFn cells were infected with wtAAV2 (MOI 20'000). At 24 hpi, the cells were fixed and processed for multicolor IF analysis combined with FISH, CLSM. (A) Intact capsids (green) or capsid proteins (yellow) were detected using either an antibody against intact AAV2 capsids (conformational capsid epitope) or an antibody (linear epitope) against VP1, VP2 and VP3. AAV2 DNA (magenta) was detected with an Alexa Fluor (AF) 647 labeled, amine-modified DNA probe that binds to the AAV2 genome. Nuclei were counterstained with DAPI and illustrated as white lines. The wtAAV2 infected cells shown were assigned as pattern I and II according to the intensity of the capsid staining and following a similar classification as defined in Fig 4. (B) Quantification of at least 50 nuclei positive for intact AAV2 capsids or capsid proteins during cell cycle progression.
(TIFF)

**S14 Fig. Helper virus-supported AAV2 DNA replication occurs in nuclear replication compartments that are distinctly separate from nucleoli.** NHF cells were mock-infected or infected with wtAAV2 (MOI 10'000), HSV-1 (MOI 0.5) or co-infected with wtAAV2 (MOI 5'000) and HSV-1 (MOI 0.5). At 12 hpi, the cells were fixed and processed for multicolor IF analysis combined with FISH and CLSM. Nucleoli (Nuc) were visualized using an antibody against fibrillarin (yellow). wtAAV2 replication compartments were stained using a primary antibody specific for the AAV2 Rep proteins (green). AAV2 DNA (red) was detected with an Alexa Fluor (AF) 647 labeled, amine-modified DNA probe that binds to the AAV2 genome. Nuclei were counterstained with DAPI and illustrated as blue lines.
(TIF)

**S1 Movie. Maximum intensity projection of AAV2 genome positive nucleoli.** NHF cells were infected with wtAAV2 (MOI 20'000). At 24 hpi, the cells were fixed and processed for multicolor IF analysis combined with FISH and CLSM. Nucleoli were visualized using an antibody against fibrillarin (yellow). Intact capsids were stained using an antibody that detects a conformational capsid epitope (green). AAV2 DNA (red) was detected with an Alexa Fluor (AF) 647 labeled, amine-modified DNA probe that binds to the AAV2 genome. Reconstructions were generated using Imaris V.9.6.
(MP4)

## Author Contributions

**Conceptualization:** Sereina O. Sutter, Hildegard Büning, Cornel Fraefel.

**Data curation:** Sereina O. Sutter, Anouk Lkharrazi, Kevin Michaelsen, Anita Felicitas Meier.

**Formal analysis:** Sereina O. Sutter, Anouk Lkharrazi, Kevin Michaelsen.

**Funding acquisition:** Cornel Fraefel.

**Investigation:** Sereina O. Sutter, Anouk Lkharrazi, Elisabeth M. Schraner, Kevin Michaelsen, Jennifer Marx, Bernd Vogt.

**Methodology:** Sereina O. Sutter, Anita Felicitas Meier, Cornel Fraefel.

**Project administration:** Sereina O. Sutter, Cornel Fraefel.

**Resources:** Bernd Vogt, Hildegard Büning, Cornel Fraefel.

**Software:** Sereina O. Sutter, Anouk Lkharrazi, Kevin Michaelsen.

**Supervision:** Cornel Fraefel.

**Validation:** Sereina O. Sutter, Anouk Lkharrazi, Anita Felicitas Meier.

**Visualization:** Sereina O. Sutter, Anouk Lkharrazi, Elisabeth M. Schraner, Kevin Michaelsen.

**Writing – original draft:** Sereina O. Sutter.

**Writing – review & editing:** Sereina O. Sutter, Anouk Lkharrazi, Elisabeth M. Schraner, Anita Felicitas Meier, Hildegard Büning, Cornel Fraefel.

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
