## [Decision Letter · Decision Letter 0]

11 Jan 2022

Dear Dr. Fraefel,

Thank you very much for submitting your manuscript "Adeno-associated virus type 2 (AAV2) uncoating is a stepwise process and is linked to structural reorganization of the nucleolus" for consideration at PLOS Pathogens. As with all papers reviewed by the journal, your manuscript was reviewed by members of the editorial board and by several independent reviewers. In light of the reviews (below this email), we would like to invite the resubmission of a significantly-revised version that takes into account the reviewers' comments. 

We cannot make any decision about publication until we have seen the revised manuscript and your response to the reviewers' comments. Your revised manuscript is also likely to be sent to reviewers for further evaluation.

Sincerely,

Matthew D Weitzman, Ph.D.

Guest Editor

PLOS Pathogens

Karl Münger

Section Editor

PLOS Pathogens

Kasturi Haldar

Editor-in-Chief

PLOS Pathogens

orcid.org/0000-0001-5065-158X

Michael Malim

Editor-in-Chief

PLOS Pathogens

orcid.org/0000-0002-7699-2064

Editors' comments: The reviewers appreciated the technical approach taken and the importance of the question being addressed.  However, they raise technical concerns, and highlight the descriptive nature of the study that lacks functional links for virus infection and vector transduction.  All comments raised by these thoughtful reviews should be addressed. 

Reviewer's Responses to Questions

**Part I - Summary**

Reviewer #1: Since the first observations of the presence of AAV capsids within the nucleolus, several studies have shown this nuclear body interacts with AAV at both early and late steps of the viral life cycle, a notion reinforced by finding that AAV capsids interact with nucleolin and nucleophosmin (B23). However, while the nucleolus is believed to favor capsid assembly during the late stage of infection, its function during the early infectious steps is still unclear. In particular, previous studies conducted with recombinant AAV vectors have suggested that the nucleolus may sequester AAV capsids upon nuclear entry and that capsid disassembly and genome uncoating requires their translocation in the nucleoplasm.

The aim of this study was to revisit this notion using immuno-FISH analyses to detect the release of the wt AAV2 genome together with the viral capsids and/or VP proteins upon infection of human primary fibroblasts. The results presented by Sutter and colleagues show that wt AAV2 single-stranded DNA can be detected in the nucleolus of the infected cells either associated to assembled AAV2 capsids or AAV2 VP proteins. Further, the authors show data that remodeling of the nucleolar structure, a phenomenon known to occur upon cell division, favors capsid disruption and release of AAV2 DNA. They conclude from these data that uncoating of AAV2 occurs in the nucleolus upon its structural reorganization during cell cycle.

While the analyses performed by Immuno-FISH are well-conducted and informative, the whole study suffers from the lack of functional analyses to support the conclusion. In particular, the absence of information about the functionality of the viral DNA released into the nucleolus, in terms of capacity to establish as a stable episome and to initiate viral replication makes any conclusion about this process premature.

Reviewer #2: Sutter et al., have provided a body of data supporting a stepwise manner of AAV uncoating and linked to structural reorganization of the nucleolus. Utilizing immunofluorescence analysis and in situ hybridization the authors attempt to tract and segregate the AAV capsid vs AAV DNA genome after infection in vitro. It is very well written, and reads like an unraveling detective story, where the authors are following the clues (data that emerged from one experiment) to inform additional experiments to refine and expand their hypothesis. The strength and the novelty of the studies reside in the author’s ability to follow the process of AAV2 infection and uncoating at a single cell resolution, therefore providing a unique glimpse into AAV2 biology. A potential weakness is the number of particles utilized in order to obtain a convincing signal for imaging. It would be prudent if the authors could compare the doses utilized to capture the data in their experiments with published data studying the infectious MOI required to obtain transduction. The potential risk of saturating pathways with virus particles in order to observe an image should be discussed. Regardless, the authors performed robust experiments, and provide conservative interpretation.

One major missing piece is the lack of discussion on how those observations affect the utility of AAV vectors for gene therapy medicines. AAV vectors are used widely to develop gene therapy vectors for where drug potency is dependent on the ability of the vectors to transduce non-dividing cells in a host. One suggestion is to include a discussion on how these data may be relevant for our understanding of how gene therapy works/may be improved. This work was done using wild type AAV2. Are those observations specific only to wild type AAV2? The authors should comment on this point. Do the authors expect that recombinant AAV vectors may behave differently? Please address in discussion. Do the authors expect that different AAV serotypes will be similar to the wild type AAV2? If such discussion is included, this position the manuscript to be of broader interest.

Minor concern

Reference 25 is repeated as ref 28. It appears the authors have inadvertently used the same ref with two different numbers, either remove and change the numbering or replace with correct reference.

Reviewer #3: The study by Sutter et al is focused on dissecting capsid uncoating steps in the infectious pathway of adeno-associated virus serotype 2 in mammalian cells. This step is critical for subsequent events in the AAV life cycle such as second strand synthesis and transcription. The study is technically sound and the authors skillfully utilize fluorescence imaging techniques and quantitation to dissect subtle shifts in AAV biology under different pharmacologically manipulated host cell conditions. Review of the existing literature and data interpretation are woven together eloquently and the overall manuscript is well written. Major strengths lie in the technical execution and quality of the study. Significant weaknesses are identified with regard to study parameters and overall descriptive nature of the findings. No significant new mechanistic insight is gleaned from the data. Major concerns include:

1. The use of fibroblasts as a benchmark for studying AAV biology is a concern. In the gene therapy context, it is well understood that most AAV serotypes suffer significant bottlenecks in the infectious pathway(s) pertinent to fibroblasts (e.g. inefficient endosomal trafficking pathways and proteasomal degradation). It is unlikely that the findings are pertinent to AAV capsid biology in general or in the context of pertinent cells such as neurons, cardiomyocytes or hepatocytes as examples.

2. The immunofluorescence based image analysis raises several significant concerns. First, the A20 antibody against the intact AAV2 capsid does not appear to recognize the capsid following conformational changes upon nuclear entry (e.g., multiple J Virol studies from Kleinschmidt, Asokan, Muzyczka and other groups). Thus the lack of A20 signal is not indicative of fully uncoated capsids. In addition, the A1/A69 antibodies against the VP1/2 domains exhibit high levels of background/cross-reactive staining in IF studies and are more reliable for western blot analysis. Thus, the use of fluorescently labeled AAV capsids is generally more reliable (e.g., Xiao and Samulski, Johnson and Samulski). In particular, the ability to recover fully infectious AAV particles purified from nucleoli post-infection has been reported (Johnson and Samulski, 2009).

3. Several key steps identified such as the need for endosomal escape, effects of BafA1 on AAV2 infection have been extensively mapped by other groups, these findings do not provide any new/additional insight. Further, the impact of proteasomal inhibitors, hydroxyurea etc on the fate of AAV in the nucleus have been reported in significant detail (e.g., Johnson and Samulski). This is especially critical for the authors to consider since drugs such as Actinomycin D disrupt nucleolar organization, while proteasome inhibitors leave the nucleolus intact (both can be utilized to block transcription). Yet, both appear to result in increased infectivity/transduction. This dichotomy needs to be resolved.

4. Lastly, while the findings related to cell cycle impact on AAV-host nucleus interactions are interesting, little mechanistic insight can be gleaned from these findings. Are there nucleolar factors that mediate AAV capsid uncoating? For instance, Nicolson and Samulski show the need for nuclear import factors, while Johnson and Samulski show that similar to effects observed with proteasome inhibition, siRNA-mediated knockdown of nucleophosmin potentiated nucleolar accumulation and increased transduction 5- to 15-fold. Parallel to effects from hydroxyurea, knockdown of nucleolin mobilized capsids to the nucleoplasm and increased transduction 10- to 30-fold. A somewhat related concern that follows is that since wtAAV is being utilized, the authors do not provide transduction data. This is a critical missing piece of mechanistic data, since it is unclear whether disrupting the nucleolus/cell cycle influences uncoating and consequently, infectivity/transduction. For instance, how would the authors deconvolute the interplay between Thymidine and ActD treatments influence infectivity/transduction given ActD mediates both nucleolar reorganization and blocks transcription?

Reviewer #4: The manuscript by Sutter et al. addresses the question how Adeno-associated virus type 2 (AAV2) fully uncoat and liberate their genome in the nucleus of infected cells. This is an important and timely topic because AAVs are increasingly and successfully used as gene therapy vectors. Knowledge that improves our understanding of AAV transduction thus may influence vector development or application. In addition, how virus uncoating is achieved upon cell entry is still not very well known.

The presented work uses quantitative single cell imaging analysis to investigate the subcellular distribution of intact AVV2 capsids, capsid proteins (for disassembled virions) and AVV2 genomes (using FISH analysis for free or partially exposed genomes) over time and under different physiological conditions. They find that AAV2 virions partially exposing the viral genome can be found in the cytosol and accumulate over time in the nucleolus. They further show that the nucleolus is also the place where AAV2 virions fully uncoat, probably in a manner that involves structural and/or functional reorganization of the nucleolus.

This work is technically well executed and the data presentation is mostly clear. However, the conclusions that can be drawn are limited by the descriptive nature of the data and I struggle at times to follow the authors line of argument. The uncoating mechanism in the nucleolus is not addressed.

I have the following comments;

1- The quantification in Fig. 1B/C indicates that the ratio of partially uncoated particles (capsid+DNA+) increases in the cytosol as well as in the nucleus over time. How was this ratio determined as I find it hard to identify individual capsids/DNA signals especially in the nucleolus at later time points?

2- The authors argue that their data show a partial disassembly step taking place in the cytosol (line 473-476), which requires acidification (Fig3). Is the bafilomycin treatment not rather affecting AAV2 uptake/endosome penetration/escape (bafilomycin treated cells seem to have less overall cell associated virions in the examples shown) ? I am also surprised to see that the endosome specific 76HD/AN mutant shows large scale partial disassembly (Fig. S1), which in my opinion argues more for an intra endosomal/endsome penetration associated partial disassembly. Could you please comment why you think this is a cytosolic step? Please provide experimental evidence for a cytosolic localization of the partial disassembled virions.

3- How are the two observed virion pools (capsid+DNA+ vs. capsid+DNA-) in the two compartments (nucleus and cytosol) functionally connected? I wonder if the partially disassembled pool in the cytosol contributes to the nucleolar pool. If this is the case are the authors suggesting that both, partially disassembled virions (capsid+DNA+) and intact virions (capsid+DNA-) can be nuclear import substrates as both forms are present in the cytosol as well as in the nucleolus? Would this not imply that partial disassembly can also take place in the nucleolus, especially in light of their findings that all virions fully uncoat in the nucleolus ?

4- The observation of some cells displaying nucleoli full of genomes but devoid of virions (Fig.4) and of cells with VP1/2/3 specific signals but absence of virion signals (Fig.5) is intriguing. Fig4 and 5 should be redone as time point experiment in cycling cells and the phenotypes should be quantified. One would expect that if the phenotypes I-III are consecutive steps that this is reflected over time and the number of capsids should diminish and the VP1/2/3 signal should increase. A similar kinetic in resting cells or contact inhibited cells should result in an uncoating defect.

5- Fig. 8 Why is there so much more DNA in the cytosol when using ActD (Fig.8A) ? Would that also be the case in absence of the Thymidine block ? How was the nucleolar area determined if ActD treatment redistributes the marker ?

**Part II – Major Issues: Key Experiments Required for Acceptance**

Reviewer #1: -Fig.1: the observations by immuno-FISH should be correlated with a kinetic analysis of the accumulation of viral DNA (total and packaged) as well as the capsids in the cytoplasm/nucleoplasm and nucleolus after cell fractionation . It would be also important to know if any viral RNA can be detected. Indeed, even if wt AAV2 cannot replicate in the absence of helper virus, a low level of Rep RNA may be produced, at least at the onset of infection.

- Fig.1: several AAV DNA-positive/capsid-negative spots are visible in the cytoplasm of the cells. Are these signals still visible after treatment of cells with nucleases (DNase or RNAse)? And how does this treatment affect the capsid-positive/DNA-positive signals?

- Fig.4: does the capsid-negative/AAV DNA-positive signal increase over time?

- Release of viral DNA could occur by a physiological uncoating process or following disruption of the capsids, both mechanisms not necessarily resulting in the same outcome in terms of viral establishment. Does the capsid-negative nucleolar AAV2 DNA signal correlate with the formation of double-stranded circular episomes?

- Proteasome inhibitors have been shown to increase the accumulation of capsids in the nucleolus as well as transduction with rAAV vectors. What is the effect of this treatment on the FISH signals detected in the nucleolus?

- Similarly, previous studies haves shown that siRNA against nucleolin and B23 alter the translocation of the capsids from the nucleolus to nucleoplasm and transduction with AAV vectors. How do these treatments affect the presence of capsid-negative AAV2 DNA in the nucleolus and the formation of double-stranded circular episomes?

- Since the authors are using primary fibroblast, growth arrested cells should be used instead of artificially-arrested cells to examine the fate of the capsids/VP proteins and viral DNA in the nucleolus of non-dividing cells. This would also allow to keep the cells in culture for longer periods to perform kinetic analyses.

Reviewer #2: No major issues

Reviewer #3: N/A

Reviewer #4: - Please provide experimental evidence that partial disassembly of virions (capsid+DNA+) occurs in the cytosol or change the statement

- Experiments in Fig4 and 5 should be redone as time point experiment in cycling cells and the phenotypes should be quantified.

**Part III – Minor Issues: Editorial and Data Presentation Modifications**

Reviewer #1: - Lines 82/94/455 References to previous studies showing accumulation of AAV capsid in the nucleolus are not appropriate. The references cited by the authors (Ref.14 and 15) refer to studies showing the interaction of the capsid with nucleolin and B23. But several previous studies had documented the presence of capsid in the nucleolus.

- Fig.3: why inhibition with Bafilomycin is not complete?

- Fig.3C and line 199: nuclear or nucleolar AAV2 import or counts? Why counting only capsid-positive signals and not, as before capsid-positive/DNA-positive signals?

- Fig8A: actinomycin D treatment seems to equally increase the AAV2 DNA signal in cytoplasm. How do the authors explain this? Actinomycin D can inhibit the intracellular transport of viral capsids as shown for Influenza virus. Could this explain the lower AAV2 capsid signal observed in the nucleolus?

-Fig.8A: despite the low level of fibrillarin staining, nucleoli are still visible by DAPI staining. Please explain.

- Line 473: uncoating or just remodeling? Extrusion of the VP1/2 N-ter may provide access to small DNA probes or simply labeled nucleotides.

- FigS2 and line 250: a negative correlation between capsid (A20) signal and individual VP staining is not evident in this image. Please explain.

- Fig.S5: decrease of cells in G1 and/or increase in G2 upon nocodazole treatment is much less evident by flow cytometry than CLSM. Please explain.

- line 529: The statement that AAV vectors do not efficiently transduce non-dividing cells is not correct. Transduction of non-dividing by AAV vectors can efficiently occurs even in vitro (provided that the receptor is present) but with a different kinetics than that observed on dividing cells.

Reviewer #2: No minor issues

Reviewer #3: N/A

Reviewer #4: - please note in each legend if the images are confocal midsections or if they are Z-projections. An outline of the nucleus in addition to the nucleolus might help. This is important to appreciate the localization of objects inside/on-top or below of the nucleus.

- Please make sure that Figure numbers are mentioned in the text in the order they appear (e.g. Fig 1C, Fig.4C appear out of order)

- Fig9 please label what is shown in top vs bottom line (same for FigS9), graph and graph legend don’t match (light color for VP, no light grey color in graph) (same for FigS9)

PLOS authors have the option to publish the peer review history of their article (what does this mean?). If published, this will include your full peer review and any attached files.

Reviewer #1: No

Reviewer #2: **Yes: **R Jude Samulski PhD

Reviewer #3: No

Reviewer #4: No
---

## [Decision Letter · Decision Letter 1]

8 Jun 2022

Dear Dr. Fraefel,

We are pleased to inform you that your manuscript 'Adeno-associated virus type 2 (AAV2) uncoating is a stepwise process and is linked to structural reorganization of the nucleolus' has been provisionally accepted for publication in PLOS Pathogens.

Best regards,

Matthew D Weitzman, Ph.D.

Guest Editor

PLOS Pathogens

Karl Münger

Section Editor

PLOS Pathogens

Kasturi Haldar

Editor-in-Chief

PLOS Pathogens

orcid.org/0000-0001-5065-158X

Michael Malim

Editor-in-Chief

PLOS Pathogens

orcid.org/0000-0002-7699-2064

The reviewers all appreciated the attention and modifications incorporated into the revised manuscript. I encourage the authors to attend to the additional minor concerns raised by Reviewers 1 and 3 when they submit the final version of this manuscript in order to clarify even further the data and interpretations.

Reviewer Comments (if any, and for reference):

Reviewer's Responses to Questions

**Part I - Summary**

Reviewer #1: Altogether, this work convincingly shows that AAV2 capsids massively accumulate in the nucleolus, where viral/vector DNA can be released and detected in high amounts, and that, in cycling cells, release of viral/vector DNA can be enhanced following disruption of the nucleolus occurring during cell division. What remain unclear is the fate of this nucleolar DNA and its links with the establishment of functional episomes. In other words, is unpackaged nucleolar AAV DNA solely responsible for the establishment of the infection, presumably after delocalization to the nucleoplasm, or are functional episomes produced by other less abundant uncoating events occurring outside of the nucleolus? The conclusion that “complete” uncoating of the AAV genome occurs exclusively in the nucleolus and that this process requires cell division completely disregards other possible interpretations and hypotheses that cannot be excluded in the absence of additional functional data (presence of circular episomes, RNA etc..).

Reviewer #2: Authors addressed all comments.

Reviewer #3: This revision provides compelling evidence of nucleolar reorganization correlating with AAV uncoating/genome release. The authors should be commended for the rigor and depth of experiments conducted to dissect the aforementioned phenomena. That being said, several gaps remain in tying the observations together. In particular, the authors could consider addressing the following. (1) the observed signals in endosome could simply arise from capsids being subjected to protease degradation and hence on a "dead end" path. Blocking with BafA1 also alters endosomal trafficking and cannot be necessarily be attributed to the beginning of uncoating in the endosome - overall the notion that uncoating begins in the endosome appears somewhat counterintuitive for a DNA virus that needs to deliver its genome to the nucleus. (2) the role of Golgi accumulation of AAV particles (MTOC) prior to nuclear entry is unaddressed. This has now been implicated as critical path for transduction by several groups and the connection or lack thereof to cell cycle/uncoating has not been discussed (one suggestion is that the manuscript might benefit by restricting the discussion to post-nuclear entry events rather than post-uptake) and (3) what are implications for non-dividing, terminally differentiated cells such as neurons? AAV clearly transduces these cell types with high efficiency in vivo - it is unclear how cell cycle mediated nucleolar reorganization might play a role in uncoating in these scenarios.

Reviewer #4: The authors have done a careful revision addressing all my concerns and answered my questions. The new time course data (fig. S12) further strengthen their initial conclusions. The revised version reads very nice and the experiments are well executed. I have no further requests.

**Part II – Major Issues: Key Experiments Required for Acceptance**

Reviewer #1: Two points deserve to be clearly discussed:

1. While it is evident that a massive accumulation of capsids and/or DNA occurs in the nucleolus, in several images, DNA+/capsid+or- signals are also visible in the nucleoplasm. For instance, in Fig.1, capsid+/DNA+ signals are visible in the nucleoplasm starting from 3 h pi followed by capsid-/DNA+ signal at 24h pi. Similar signals are visible in FigS2, FigS4, Fig.4 (panel I). The same observation can be made regarding the presence of non-assembled VP proteins and AAV DNA (Fig.6). The presence of these signals is never mentioned nor measured in the whole manuscript. While I understand that such low signals may be difficult to quantify, their presence suggests two possible hypotheses: either viral/vector DNA is uniquely uncoated in the nucleolus to subsequently move/leak into the nucleoplasm, or uncoating can also independently occur in the nucleoplasm, raising the fundamental question of which pathway (nucleolar our nucleoplasmic) leads to the formation of functional viral/vector episomes. The efforts made by the authors to show the functionality of the “nucleolar pool of DNA (Figure 5 and S6, respectively) does not, in my opinion, convincingly answer this question. For instance, the linear ds form observed at 48 pi by Southern blot (Fig.5) and following transfection (FigS6), could result either from second-strand synthesis (as claimed but not demonstrated by the authors) or simply from annealing of ssDNA genomes, a phenomenon that could be enhanced by the high concentration of ss genomes in the nucleolus and that can also “artificially” occur after isolation of ssAAV DNA. In addition, even if other forms than ssDNA are not visible in the DNA pool extracted from the nucleoplasm at 48h pi, this does not exclude that they may arise following a longer kinetics.

2. The observation that capsid disassembly is enhanced following disruption of the nucleolus is certainly relevant for cycling cells but how does this explain the capacity of rAAV vectors to transduce terminally differentiated, and therefore growth arrested, cells? If nucleolar disassembly is absolutely required for transduction then AAV vectors should be unable to transduce cells such as neurons, myotubes, etc…Here again, an hypothesis could be that in such cells uncoating may also occur in the nucleoplasm with a different kinetics of release that does not depend on nucleolar disintegration.

Reviewer #2: (No Response)

Reviewer #3: No further experiments are required. The authors efforts to address this reviewer's concerns are deeply appreciated.

Reviewer #4: NA

**Part III – Minor Issues: Editorial and Data Presentation Modifications**

Reviewer #1: - The authors frequently use the term of “complete uncoating” (see for example lines 385 and 413) without specifying that most of the analyses concerns only the nucleolus.

- Fig.2A were the virions processed exactly as cells (including the permeabilization step?)

-Fig5. The Southern blot suggests the amount of sssDNA in the nucleolus and the nucleoplasm are similar at 48H pt. Is this also visible by FISH?

- FigS5C. WB: Is this the nuclear or nucleoplasmic fraction? A marker specific of the nucleoplasmic fraction should be used to show the efficacy of nucleoli purification.

- FigS12A: please provide evidence that the cells are growth-arrested.

- Fig.8B and C, Fig.10B, FigS12B and C: please indicate on the graphs if these measures concern the whole nucleus or exclusively the nucleolus

Reviewer #2: (No Response)

Reviewer #3: N/A

Reviewer #4: NA

PLOS authors have the option to publish the peer review history of their article (what does this mean?). If published, this will include your full peer review and any attached files.

Reviewer #1: No

Reviewer #2: No

Reviewer #3: No

Reviewer #4: No

---

## [Editor Report · Acceptance letter]

1 Jul 2022

Dear Dr. Fraefel,

We are delighted to inform you that your manuscript, "Adeno-associated virus type 2 (AAV2) uncoating is a stepwise process and is linked to structural reorganization of the nucleolus," has been formally accepted for publication in PLOS Pathogens.

Best regards,

Kasturi Haldar

Editor-in-Chief

PLOS Pathogens

orcid.org/0000-0001-5065-158X

Michael Malim

Editor-in-Chief

PLOS Pathogens

orcid.org/0000-0002-7699-2064